# Variation in surface protein expression leads to heterogeneous *Trypanosoma cruzi* populations during host cell infection

Lissa Cruz-Saavedra[1], Mira Loock [1], Luiza Berenguer Antunes[1] & Igor Cestari [1,2] ✉

*Trypanosoma cruzi* possesses hundreds of genes associated with pathogenesis. The extent and organization of this diverse gene repertoire, expression, and role in infection remain unclear. Using accurate long-read sequencing and chromatin conformation capture, we assembled *T. cruzi* Sylvio X10 strain chromosomes from telomere-to-telomere. The genome provides accurate organization of multigene family genes, confirming their distribution in expanded clusters or scattered throughout the chromosomes. Quantitative proteomics shows stage-specific proteins and numerous trans-sialidases upregulated in trypomastigotes. The expression of virulence gene families varied in trypomastigotes after each round of cell infection, resulting in heterogeneous parasite populations with variable cell invasion capacity. A *T. cruzi* genome-wide yeast surface display screen against Chagas disease patients' antibodies reveals genes expressed during human infections. However, limited conservation in their antibody-binding sites suggests their sequence diversity and variation might help parasites avert antibody recognition. The data point to a role for some multigene families in infection persistence.

Many organisms evolved large gene families for specialized survival functions or to adapt to environmental changes. This is the case for odorant receptors in mammals[1], ubiquitin-like proteins in some bacteria[2], or pathogens' variant surface antigens for immune evasion[3]. Their expansion results from gene duplication and recombination events, with their chromosome organization typically influencing their expression[4,5]. The pathogen *Trypanosoma cruzi*, a single-celled protozoan parasite, has six prominent multigene family (MGF) groups encoding dozens to hundreds of genes, including trans-sialidases (TS), mucins, mucin-associated surface proteins (MASPs), dispersed gene family 1 (DGF-1), retrotransposon hot spot (RHS) and the zinc metalloprotease gp63 (GP63), accounting for ~20% of the genes in the genome[6–8]. MGFs such as TSs, mucins, MASPs are associated with *T. cruzi* pathogenesis[9] and thus are known virulence factors, whereas DGF-1 and RHS functions remain unclear. The MGFs function and control of expression are poorly understood, partly due to their repetitive and highly variable sequences, complex genome organization, and diversity among strains.

*T. cruzi* causes Chagas disease, affecting ~8 million people primarily in South and Central America, but has spread to the United States, Canada, Europe, and Asia[10]. A reduviid insect vector transmits the parasite from small mammal reservoirs to humans; however, it can also spread through congenital transmission, blood transfusion, or food contamination. In the insect, the parasite replicates as epimastigotes, differentiating into non-replicative metacyclic trypomastigotes. Metacyclics are released by the insect during a blood meal and infect the host cells. Inside the host cells, they differentiate into replicative amastigote forms. After several rounds of cell division, amastigotes differentiate into non-dividing trypomastigotes, which can spread in the bloodstream and infect new tissues, where they differentiate into amastigotes and replicate. The insect can take trypomastigotes during a blood meal and re-initiate the cycle. Chagas disease has a short acute

[1]Institute of Parasitology, McGill University, Sainte-Anne-de-Bellevue, QC, Canada. [2]Division of Clinical and Translational Research, Department of Medicine, McGill University, Montreal, QC, Canada. ✉e-mail: igor.cestari@mcgill.ca

stage, characterized by detectable parasitemia, and can evolve into a decades-long chronic disease with parasites persisting in tissues[11,12]. The host immune response against *T. cruzi* entails parasite-specific antibody-mediated attacks and CD8[+]- and CD4[+]-T cell responses[13–15]. While the immune response controls parasitemia, it does not eliminate parasites that persist in certain tissues. Persistence has been associated with the failure of CD8[+]-T cells to clear tissue infection[14,16,17] and the expression of parasite molecules that counteract the tissue immune response[18–21]. This leads to chronic disease, with approximately ~30% of cases progressing to cardiac or gastrointestinal complications, often resulting in patient disability or death[22]. There is no vaccine for Chagas disease, and nifurtimox and benznidazole are drugs used for treatment, but they are mainly effective in the acute stage[23].

The mechanisms underlying *T. cruzi*'s ability to infect multiple tissues and cause persistent infections are unclear. *T. cruzi* surface is heterogeneous, with multiple glycosylated proteins such as TSs, mucins, and MASPs usually attached to the cell surface by glycosyl-phosphatidylinositol anchors, and some change throughout the cell cycle[24]. Some of these proteins are virulence surface factors involved in immune evasion and host cell infection, and others are highly immunogenic and shed in the host bloodstream[21,25–29]. Active TSs, for example, transfer host sialic acid to the parasite surface[29,30], contributing to complement evasion[31], cell invasion[21] and cell egress[21]. Some enzymatically inactive TSs and mucins play a role in host cell receptor recognition and invasion[32,33], modulation of host cell apoptosis[34] and the immune system responses[35,36]. It is still unclear how the surface protein repertoire of *T. cruzi* changes in various developmental stages and throughout infection. Given the extensive repertoire and amino acid sequence diversity of virulence MGF genes (i.e., TSs, mucins, MASPs, GP63), a plausible conjecture is that variation in the genes expressed helps the parasite invade multiple tissues or evade host immune responses, contributing to infection persistence.

In this work, we aimed to investigate the genomic organization and pattern of expression of MGF genes in *T. cruzi*, with a particular focus on whether temporal variation in gene expression could diversify the parasite population. Moreover, we investigate how their variation affects host cell infection and host antibody interactions. Understanding the expression of MGFs in *T. cruzi* and their host interaction might provide insights into parasite infection and persistence. We generated a high-resolution genome of the *T. cruzi* Sylvio X10 strain, assembling complete chromosomes. The assembled genome highlights the extent and locations of MGF genes, typically organized as gene clusters, also described as disruptive compartments, or spread throughout the chromosomes. Quantitative proteomics revealed stage-specific MGF subsets with an upregulation of TSs in trypomastigotes. A temporal nanopore-based MGF-seq and proteomic analysis in trypomastigotes revealed expression variation of MGFs with changes occurring after host cell infection. The changes concurred with variable rates of *T. cruzi* invasion in multiple host cell types. A *T. cruzi* genome-wide yeast surface display (YSD) screen against antibodies from Chagas disease patients identified MGFs expressed during human infections. Their antibody-recognition sites exhibited limited conservation, indicating that the sequence diversity and changes in expression of some MGF members (e.g., TSs, mucins, MASPs) may contribute to antibody evasion. The data confirm MGFs' genomic organization and reveal stage-specific expression and variation in trypomastigotes, indicating that sequence diversity and variation of gene families might contribute to heterogeneous parasite populations and a potential role in immune evasion.

## Results

### High-resolution genome assembly results in chromosome-level gene organization

To understand the organization and number of MGF genes in *T. cruzi*, we sequenced and de novo assembled the genome of the TcI strain

Sylvio X10. TcI is widespread from South to North America and is typically associated with cardiovascular disease[37]. The genome was assembled using PacBio HiFi sequencing and chromatin conformation capture with nanopore sequencing (Pore-C) (Fig. 1A, Supplementary Fig. 1). The assembled nuclear genome was 38.1 Mb, with an N50 of 1.30 Mb and the largest contig being 2.63 Mb. It included 13,798 genes, of which 2934 encoded MGFs (Fig. 1A, Table 1). Thirty-one chromosomes were assembled, of which 27 included end-to-end telomeres, and four had one telomere (Fig. 1A). Additionally, we resolved the previous described[7] two haplotypes for chromosome 22, namely 22a and 22b. Haplotype 22a contains a 126,891 bp insertion, which includes L1Tc elements followed by TS genes (Fig. 1A). Ploidy analysis by depth (average 397x) revealed 29 diploid chromosomes (Fig. 1B, Supplementary Fig. 2). In contrast, chromosomes 1 and 30 were haploid and triploid, respectively. The variable depth for part of chromosome 1 suggests potential extra-chromosomal elements, whereas some chromosomes might have segmental aneuploidies, e.g. chromosome 10 (Supplementary Fig. 2). The mitochondrial maxicircle genome was hexaploid, with a complete set of genes and repeat sequences (Supplementary Fig. 3). Sequence depth using nanopore sequencing (70x) corroborated chromosome copy numbers obtained by PacBio depth analysis (Supplementary Fig. 2). Repeat sequences higher than 70% were observed in chromosomes 9, 12, 29, and 31 (Fig. 1A), all rich in MGF sequences. There were 39 short-length scaffolds ranging from 6 to 68 kb with a summed size of less than a tenth of the smallest chromosome. Synteny analysis revealed that scaffolds shared 70–98% similarity with chromosomal regions, indicating they represent haplotype differences encoding primarily hypothetical proteins and a few virulence-related MGF genes (~7% of genes) (Fig. 1C, Supplementary Fig. 4, Supplementary Table 1). Nanopore and Illumina sequencing were used to validate the assembly, demonstrating increased data mapping and quality scores for this work genome compared to the previous assembly[38], with most reads mapping to the assembled chromosomes (Supplementary Fig. 5).

The chromosomes were primarily organized into core genes encoding housekeeping and hypothetical proteins, alternating with clusters of MGF genes forming disruptive compartments (Fig. 1A, D, E). There were ~70 genes per core gene cluster and ~18 genes per disruptive MGF cluster, and core genes were syntenic across multiple *T. cruzi* strains (Supplementary Fig. 6). MGFs and L1Tc-co retrotransposon accounted for ~21% of the annotated genes (Table 1). The most abundant members were the TSs (1015 genes), particularly those from groups II and V. These were followed by MASPs (528 genes), RHS (521 genes), mucins (266 genes)—with TcMucII being the most abundant group—and L1Tc-co retrotransposons (223 genes). Chromosomes 6, 9, 11, 12, 29, and 31 contained over 70% of their sequences as repeats and ~60% of the genes encoded MGFs, with notable disruptions in the organization of forward (red) and reverse (blue) strands (Fig. 1A, E, Supplementary Fig. 2). Other chromosomes, such as chromosome 2, contained primarily core genes (Fig. 1D). Mucins, MASPs, TSs, and GP63 were distributed in MGF clusters or scattered throughout multiple chromosomes (Fig. 1D, E). In contrast, 177 DGF-1 and 378 RHS genes were enriched in subtelomeric regions, whereas 44 DGF-1 and 129 RHS genes were spread throughout disruptive regions containing MGF genes in the core chromosome (Fig. 1A, D, E, Supplementary Fig. 7). This is consistent with previous observations of their enrichment in subtelomeres[7,8,39,40], but highlights that they are not exclusive to subtelomeric regions. The enrichment of some of these genes in subtelomeric regions indicates potential differences in gene expression regulation. MGF sequences typically correlated with a high GC content (~65%) and were rich in LINE/I-L1Tc retrotransposons (Fig. 1A, D, E), except for RHS sequences, which were not GC-rich (~50%) regardless of core or subtelomeric locations (Supplementary Fig. 7). The data provide a high-resolution genome assembly of *T. cruzi* Sylvio-X10 strain and insights into MGF's widespread chromosomal organization. The alternating disruptive MGF clusters and core genes likely

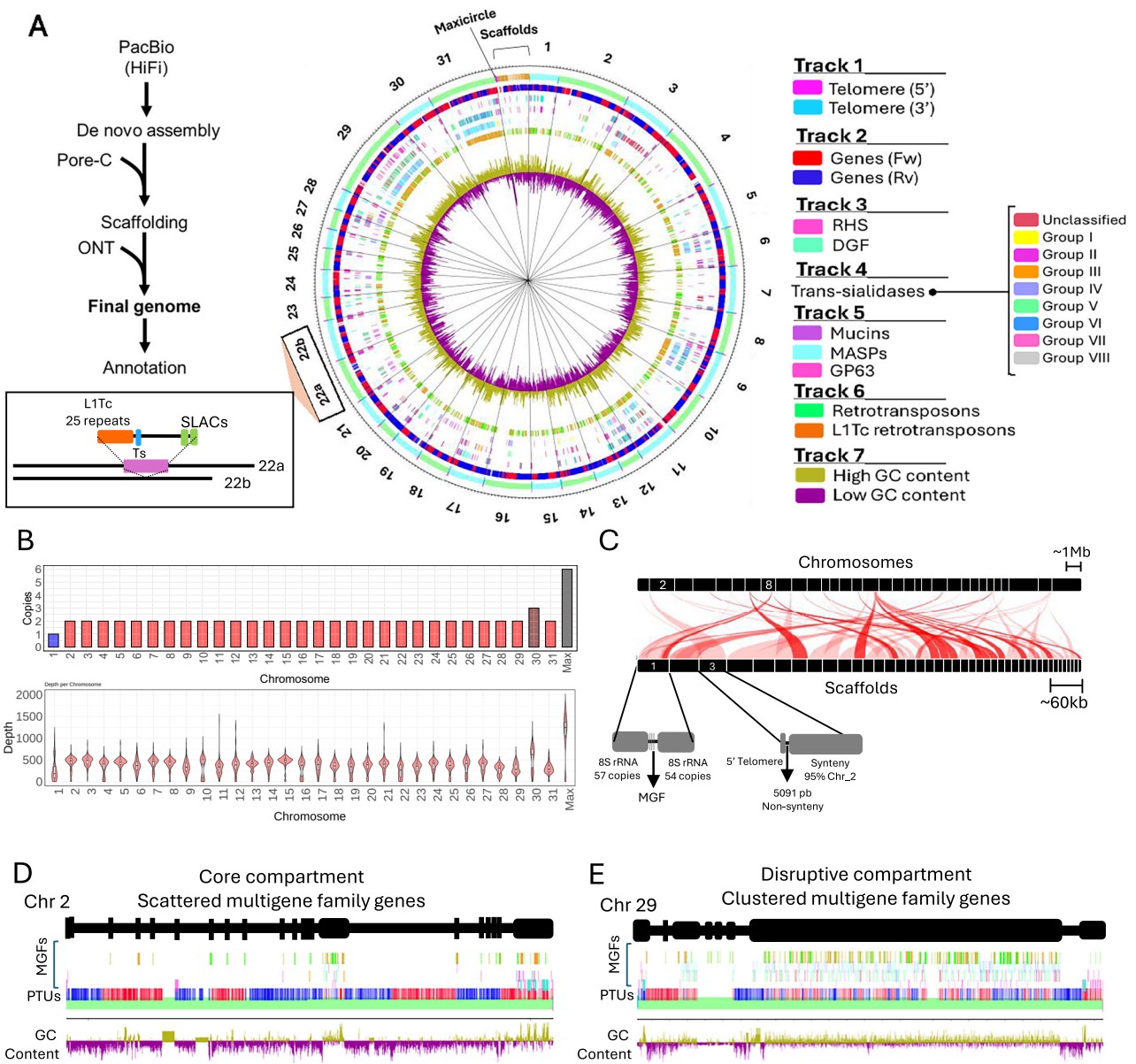

**Fig. 1 | Assembly of *T. cruzi* Sylvio X10 genome. A** The diagram (left) shows the steps to assembling the *T. cruzi* Sylvio X10 genome. The assembled genome's circular map (right) shows the distribution of genes, MGFs, telomeres, and GC content throughout the 31 chromosomes, 39 scaffolds, and maxicircles. The central circular map displays genomic features on seven concentric tracks (from outermost to innermost). Track 1 identifies telomeres (5' magenta, 3' cyan) and Track 2 shows gene orientation (forward-red, reverse-blue). Track 3 locates the RHS (pink) and DGF (green) gene families, while Track 4 maps the color-coded subgroups of TS genes. Track 5 indicates mucins (cyan), MASPs (purple), and GP63 (pink). Track 6 shows RHS (green) and the L1Tc type (orange). Track 7 shows high GC content (gold peaks) and low GC content (purple peaks) regions. The outer numbers indicate the chromosomes, with haplotypes 22a and 22b noted. The inset details differences between haplotype 22a and 22b, showing L1Tc repeats (orange), TS gene

(blue), and SLACs (green). **B** The chromosome copy number (top) and chromosome sequencing depth (bottom). In the top panel, colors represent chromosome copy number: one (blue), two (red), three (dark red), and six (gray). The violin plots illustrate the kernel density distribution of depth 10 kb mean values, while boxplots within violins summarize the distribution. The horizontal line within the box indicates the median; box limits, the 25th and 75th percentiles; whiskers, the minima and maxima within 1.5x the interquartile range; outliers beyond these limits are not shown. *n* corresponds to the number of 10 kb bins per chromosome, which varies depending on chromosome length. **C** Synteny analysis of scaffolds against all chromosomes. The diagrams show chromosome 2, a core compartment chromosome, with scattered MGF genes (**D**) and chromosome 29, a disruptive compartment chromosome, with expanded and clustered MGF genes (**E**). The diagrams follow the same color code indicated in the circular plot tracks legend in A.

reflect an evolutionary pressure to maintain MGF diversity and expression. The high-accuracy assembled genome will also help with gene expression analysis of MGF genes.

**Tandem mass tag quantitative proteomics reveals stage-specific multigene family expression**

To gain insights into the expression of MGF genes in *T. cruzi*, we performed multiplexed quantitative proteomics to determine differences

in protein expression among *T. cruzi* stages and identify expressed MGF proteins. We used tandem mass tags (TMT) to quantify proteins of each parasite life cycle stage, i.e., noninfectious epimastigotes (EP), column-purified infectious metacyclic trypomastigotes (MT), amastigotes (AM), and tissue culture-derived trypomastigotes (CT), representing the main developmental stages (Fig. 2A). Proteins were extracted, trypsin-digested, and peptides labeled with isobaric ($^{15}$N, $^{13}$C, and $^2$H) but molecularly identical tags containing a reactive group, a

**Table 1 | Distribution of annotated genes in the *T. cruzi* Sylvio X10 genome**

| Gene category | | Number of genes | Percentage (%) |
|---|---|---|---|
| Core genes (housekeeping)[a] | | 4616 | 33.45 |
| Hypothetical proteins without predicted domains[b] | | 3915 | 28.37 |
| Hypothetical proteins with InterPro domains | | 2039 | 14.78 |
| Uncharacterized proteins | | 192 | 1.39 |
| rRNAs | | 256 | 1.86 |
| tRNAs | | 74 | 0.54 |
| Trans-sialidases | Trans-sialidase unclassified | 236 | 1.71 |
| | Trans-sialidase group I | 19 | 0.14 |
| | Trans-sialidase group II | 270 | 1.96 |
| | Trans-sialidase group III | 22 | 0.16 |
| | Trans-sialidase group IV | 13 | 0.09 |
| | Trans-sialidase group V | 265 | 1.92 |
| | Trans-sialidase group VI | 105 | 0.76 |
| | Trans-sialidase group VII | 10 | 0.07 |
| | Trans-sialidase group VIII | 75 | 0.54 |
| Total trans-sialidases | | 1015 | 7.36 |
| Mucins | Mucin TcMuc | 4 | 0.03 |
| | Mucin TcMucI | 5 | 0.04 |
| | Mucin TcMucII | 182 | 1.32 |
| | Mucin TcSmugI | 24 | 0.17 |
| | Mucin TcSmugs | 35 | 0.25 |
| | Mucin-like glycoprotein | 16 | 0.12 |
| Total mucins | | 266 | 1.93 |
| Mucin-associated surface protein (MASP) | | 528 | 3.83 |
| Dispersed gene family protein 1 (DGF-1) | | 221 | 1.60 |
| Surface protease GP63 | | 160 | 1.16 |
| Retrotransposon hot spot (RHS) protein | | 521 | 3.78 |
| L1T | L1Tco protein | 165 | 1.20 |
| | L1Tc protein | 158 | 1.15 |
| Total L1T | | 223 | 1.62 |
| Total multigene family and L1Tc | | 2934 | 21.26 |
| Total annotated genes per haploid genome | | 13,161 | 95.38 |
| Total annotated genes per diploid genome | | 13,798 | 100 |

The haplotype genome, which includes chromosome 22 haplotype a, has 13,798 annotated genes. Chromosome 22 haplotype a contains an additional 637 genes compared to haplotype b.
[a]Includes all protein-coding genes except hypothetical proteins and multigene families.
[b]The total number of genes encoding hypothetical proteins is 5954.

space arm for mass normalization, and a mass reporter (from 126 to 135 Da) followed by LC-MS/MS to identify peptides and quantify the abundances of reporter ions (Fig. 2B). Principal component analysis (PCA) revealed differences among each parasite life stage and consistency within biological replicates (Fig. 2C). The dataset comprised 42,348 peptides and 5993 proteins with quantifiable differences among various parasite stages (Supplementary data 1). Hundreds of proteins in various cellular processes were up- and down-regulated throughout the life stages, defining stage-specific and common proteomes (Fig. 2D, Supplementary Fig. 8). The data included known stage-specific markers, such as GP82 and GP90 in MTs[28], TSs in CTs[29,30], amastin in AMs[41], and proteins involved in energy metabolism in EPs (Fig. 2D-F). It also shows the expression of known vaccine candidates (e.g., Tc24) and diagnostic targets (e.g., HSP70 and histones)[42] (Fig. 2E, Supplementary data 1). Analysis of the differentially expressed proteins between infectious and noninfectious forms (Fig. 2F, Supplementary Fig. 8) revealed over 600 potential virulence factors, as well as metabolic, morphological, and surface coat proteins associated with each infectious stage.

We identified a stage-specific pattern of MGF protein expression (Fig. 3A), characterized by variations in protein abundance across each stage. TSs, mucins, and MAPSs were significantly upregulated in CTs, with many TS proteins from all groups expressed (Fig. 3A, B, Supplementary Fig. 9). In contrast, DGF-1 and RHS were upregulated in AMs, MTs, and EPs, whereas GP63 were upregulated in AMs and CTs (Supplementary Fig. 9), in agreement with other observations[24,25,28,33,41,43]. Analysis of TS subsets showed enrichment of all TS groups in CTs compared to other stages (Fig. 3C, Supplementary Fig. 9). The MGF proteins expressed in CTs were encoded by genes distributed across multiple chromosomes rather than a single locus (Fig. 3D, E). However, chromosomes 6, 9, 29, and 31, enriched in MGFs, contributed to a large proportion and exhibited higher MGF abundance in CTs compared to other stages (Fig. 3D, Supplementary Fig. 10). RNA-seq and proteomic analysis confirmed the expression of large subsets of MGF genes from multiple chromosome loci (Fig. 3E, Supplementary Fig. 11). Since single-cell data show transcription of different TSs in different cells[44], it is possible that the diversity of MGF proteins detected reflects a heterogeneous parasite population. Analysis of mRNA abundance in CTs showed a bimodal distribution of TSs (Fig. 3F), distinct from other MGF and core gene transcripts (Supplementary Fig. 12), with RNA levels around 1.2 and 5 logs. The high abundance of TS mRNAs correlated with the distribution of TS proteins expressed, suggesting a threshold accumulation in mRNA levels for their translation. The data revealed a stage-specific *T. cruzi* proteome, identifying stage-specific subsets of MGF proteins and an upregulation of TSs in CTs. It also indicates that MGFs are expressed from multiple locations on various chromosomes, resulting in various MGF proteins expressed in the parasite population.

## Variation in the multigene family's expression results in heterogeneous *T. cruzi* populations

Our proteomics data showed several MGF proteins expressed in the CT population. Single-cell transcriptomics data indicated only a few MGFs expressed per parasite[44,45], suggesting diversity in MGFs expressed at the population level. Hence, we postulated that MGF gene expression varies temporally in CTs after every round of cell infection, resulting in heterogeneous parasite populations. We developed an MGF-seq approach to select all expressed MGF mRNAs and determine their expression changes in CTs. We designed sets of primers targeting the 3' conserved ends of MGF genes and the splice leader sequence (Supplementary Table 2) attached to the 5' UTRs of all trypanosome mRNAs[46,47], followed by nanopore sequencing. We infected H9-C2 cardiomyocyte cells with MTs and collected the generated CTs (Fig. 4A). Then, we performed four rounds of host cell infection with CTs accompanied by MGF-seq at each round to investigate MGF expression changes (Fig. 4A). The approach identified 2465 MGF transcripts, ~91% of all MGF genes annotated in the genome, and 2029 (~82%) transcripts varied throughout CT generations (Fig. 4B). Some transcripts appeared more often than others, although still varied in expression (Fig. 4B brackets). There were twice as many MGF genes expressed in CTs (1301 ± 436 genes) as in MTs (686 genes) (Fig. 4B, Supplementary Fig. 14), with 154 TSs expressed at any time in the CT population, representing ~15% of the TS repertoire. However, TS expression levels varied significantly and changed throughout CT

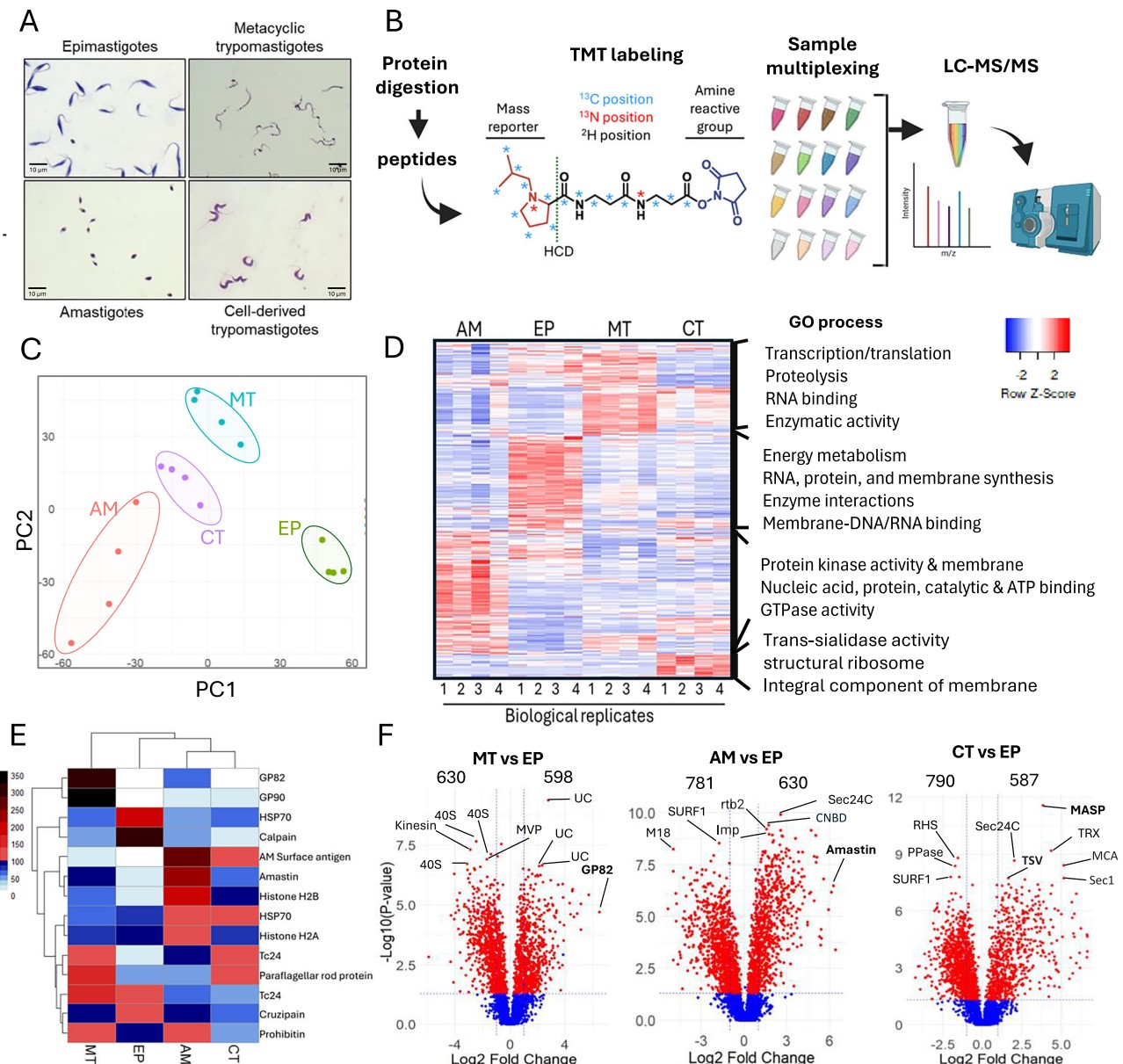

**Fig. 2 | Quantitative proteomics of *T. cruzi* life cycle stages. A** Giemsa staining of the four main stages of *T. cruzi* used for proteomics. MTs and AMs were purified using DEAE-Sephadex[61]. **B** Diagram of TMT-labeling and mass spectrometry. Proteins from each life stage were extracted (four biological replicates), trypsin digested, and peptides labeled with a 16-plex isobaric molecule with an amine-reactive group and mass reporter. After labeling, samples were mixed and co-analyzed by mass spectrometry. Created in BioRender. Cestari, I. (2025) https://BioRender.com/sm8yag9. **C** PCA plot of the mass spectrometry results for each parasite stage. Each dot represents a biological replicate. CT, cell-derived trypomastigotes (purple); AM, amastigotes (red); MT, metacyclic trypomastigotes (blue); EP, epimastigotes (green). **D** Heatmap of total protein expression change; it does not include MGFs. The most represented gene ontology (GO) processes are indicated on the right. **E** The heatmap shows the expression of selected known stage-specific proteins, vaccine candidates, and diagnostic targets. **F** Volcano plots show a comparison of protein expression between infectious stages (MT, CT, AM)

and the non-infectious stage (EP). Red dots are significant changes, i.e., fold-change ≥2 and *p*-value ≤ 0.05, whereas blue dots are non-significant. Statistical comparisons were performed using ANOVA (individual proteins) with a confidence level of 95% for four stages across four biological replicates. *P*-values were adjusted for multiple comparisons using the Benjamini-Hochberg method. See Supplementary data 1 for exact *p*-values. Stage-specific markers are indicated in bold (GP82, Amastin, MASP, TS). 40S 40S ribosomal protein S2, MVP major vault protein, UC uncharacterized protein, GP82 surface glycoprotein 82 kDa, M18 glutamamyl carboxypeptidase putative; M18 metallo-peptidase clan MH family M18, SURF1 surfeit locus protein 1-like protein, Imp importin 1, Rtb2 GTP-binding nuclear protein rtb2, Sec24C transport protein Sec24C, CNBDs cyclic nucleotide-binding domain-containing protein, PPase inorganic pyrophosphatase, RHS retrotransposon hot spo, TsV trans-sialidase group V, MASP mucin-associated surface protein, TRX thioredoxin, MCA metacaspase, Sec1 putative sec1-like protein.

generations or biological replicates, with all subgroups represented (Fig. 4B, C, Supplementary Fig. 14), indicating TS heterogeneity and variation within the parasite population. Analysis of two distinct RNA-seq confirmed multiple MGF genes expressed at different levels in the CT population (Supplementary Fig. 13), also noted by the proteomic analysis (Fig. 3). TMT-labeling proteomics over CT generations

confirmed MGF expression changes at the protein level (Fig. 4C), confirming diversity and variation in MGF expression. It is likely that individual parasites express only subsets of the MGFs and vary their expression throughout the course of infection.

To determine if the variation in MGF expression affects host cell infection, we quantified the production of CTs every round of H9-C2

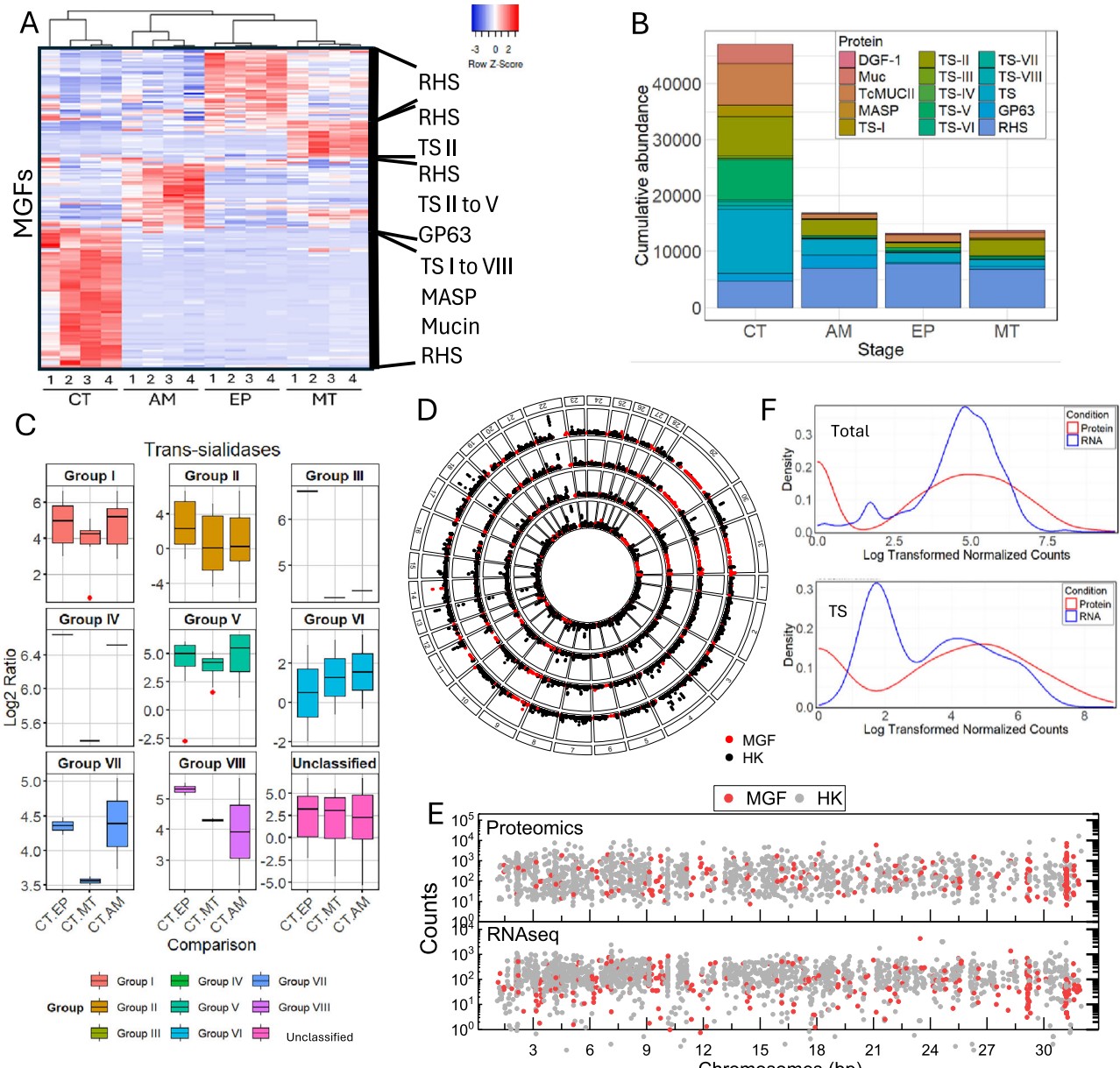

**Fig. 3 | MGF proteins are expressed from various chromosome loci. A** Heatmap of MGF protein expression by TMT-labeling proteomics. Predominant protein groups are indicated on the right. **B** Cumulative abundance of MGF proteins quantified by TMT-labeling proteomics in each life cycle stage. **C** Quantitative comparison of TS protein subgroups between CT vs EP, CT vs MT, and CT vs AM. The boxplot shows the distribution of log2 abundance ratios for different TS groups comparing *T. cruzi* life stages. The horizontal line within the box indicates the median; box limits, the 25th and 75th percentiles; whiskers, the minima and maxima within 1.5x the interquartile range; outliers are plotted as individual red points. *n* corresponds to the number of proteins within four biological replicates. See Supplementary data 1 for exact values. **D** The circular plot shows the chromosome location associated with the MGF protein expressed. MGF multigene family proteins (red); HK core housekeeping proteins (black). Data show protein expression means of four biological replicates for each parasite life stage. **E** Mean RNA and protein abundance counts in all chromosomes MGF (red) and HK (gray). **F** Distribution of total cell transcriptome (top) and TSs (bottom) in CTs. RNA distribution (blue) and protein distribution (red). The data show the results of four biological replicates.

infection. Moreover, we performed invasion assays using H9-C2 (cardiac), HeLa (epithelial), and 3T3 (fibroblast) cells with MTs or CTs obtained from H9-C2 or HeLa cells. Interestingly, there was an increase in CT production with each sequential round of H9-C2 infection (Fig. 4D). In contrast, invasion assays showed variations in the rates of host cell invasion between MTs and CTs, or amongst CT generations and host cell types (Fig. 4E). The variation in invasion rates occurred with CTs obtained from H9-C2 or HeLa cells, implying it is independent of the host cell used to generate the CTs. The data indicate that variations in *T. cruzi* surface protein expression may result in heterogeneous parasite populations after host cell infection; the variation in

proteins expressed may increase the parasite's ability to invade multiple cell types rather than selecting a specific cell type. This may be due to changes in parasite surface proteins allowing them to engage with different host cell receptors for invasion. On the other hand, the increase in CT production after consecutive rounds of infection (Fig. 4D) does not correlate with invasion rates (Fig. 4E) and thus might be related to other processes. There was a notably high rate of invasion with the first generation of CTs (G1) (Fig. 4E), which correlated with their lower diversity in MGFs expressed (Fig. 4B, Supplementary Fig. 14). This suggests a more homogeneous parasite population at G1, perhaps favoring surface protein engagement with host cell receptors

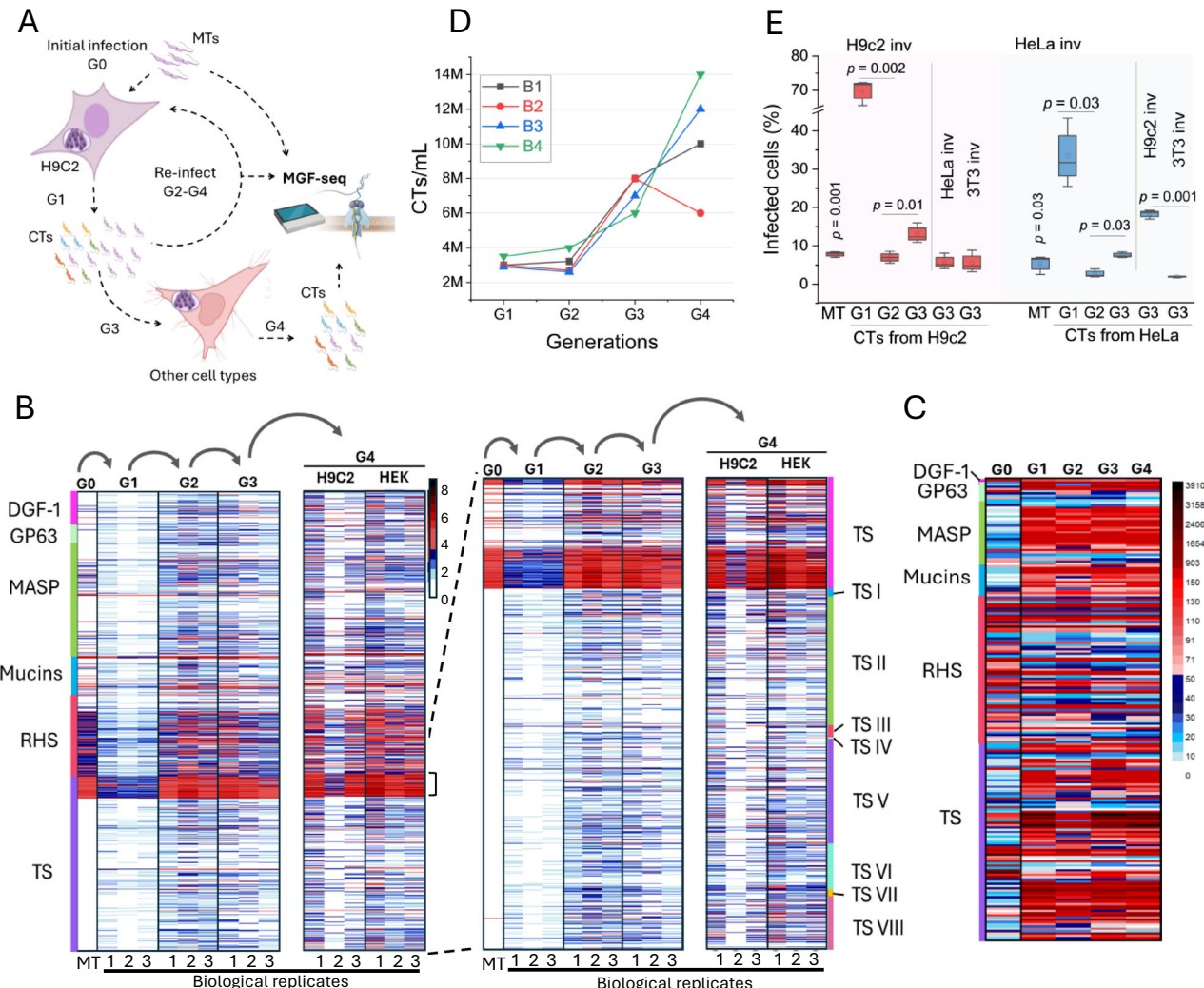

**Fig. 4 | Variation in MGF gene expression in trypomastigotes. A** Diagram of MGF-seq experimental design. H9-C2 cardiomyocyte cells were infected with MTs, and CTs were collected from the supernatant and used to re-infect H9-C2 cells. The process was repeated four times, and CTs were collected for MGF-seq each time. After the third round of infection, HEK293T cells were infected, and CTs were collected for MGF-seq. G, generation. CTs, cell-derived trypomastigotes; MTs, metacyclic trypomastigotes. Created in BioRender. Cestari, I. (2025) https://BioRender.com/937c2v0. **B** Heatmap of MGF-seq from MTs and CTs. Each round of infection (G) and biological replicate is shown. On the right is an analysis of TS expression by subgroups. Unclassified TSs are labeled TS. The heatmap shows a log2 scale of normalized read counts. The vertical line color indicator in the heatmap (left) shows disperse gene family 1 (DGF1, pink), surface glycoprotein of 63 kDa (GP63, light blue), mucin-associated surface protein (MASP, green), mucins (blue), retrotransposon hot spot (RHS, red), and trans-sialidase (TS, purple). The vertical color line in the heatmap (right) shows TS-unclassified (pink), TS-I (blue), TS-II (green), TS-III (red), TS-IV (yellow), TS-V (purple), TS-VI (light blue), TS-VII (orange),

and TS-VIII (light red). **C** Quantification of MGF proteins from MTs (G0) and CTs (G1-G4) by TMT-labeling and LC-MS/MS at multiple generations as described in (**A**). The heatmap shows a linear scale of protein relative abundance. The same identifier colors as the heatmap (left) in (**B**) were used. **D** Quantification of CTs produced and released into the host cell supernatant each round of infection from (**B**). In each round, 30,000 H9-C2 cells were infected (1:10, cells to parasites), and CTs were collected every 6 days post-infection. M millions. **E** Invasion assay with H9-C2 and HeLa, or 3T3 cells for *T. cruzi* MTs or CTs (produced at multiple generations in H9-C2 (light red shading) or HeLa cells (light blue shading). 30,000 cells were incubated with 10 parasites per cell for 3 h. The boxes in the graph display a 25–75% distribution; the horizontal line within the box represents the mean; vertical lines represent the standard deviation for three biological replicates. The p-values, indicated in the figure, show comparisons between subsequent generations and were performed using pairwise t-tests (two-sided) with a 95% confidence level based on three biological replicates.

for invasion, whereas the increased MGF diversification in subsequent generations might have affected invasion efficiency to specific cell types. The data show variation in MGF expression at both the RNA and protein levels in CTs after host cell egress, suggesting that this variation may diversify *T. cruzi* population during infection.

### Diversity and immunogenicity of MGF proteins during human infections

To gain insights into MGF protein immunogenicity and expression variation during human infections, we screened a *T. cruzi* genome-wide library for YSD[48] against antibodies from patients with Chagas disease

or healthy individuals (Fig. 5A). The library size corresponds to ~30-fold the parasite genome, including all genes, and has ~4 million clones with ~270 clones per gene[48]. The library expression is induced by galactose, resulting in parasite proteins or fragments thereof expressed on the yeast surface via the agglutinin mating system Aga2p-Aga1p[49] (Fig. 5A). The YSD library was incubated with pooled sera from chronic-stage Chagas disease patients (ChD) and compared to those from healthy individuals. Although the strains infecting the patients are unknown, MGF genes are present in all *T. cruzi* strains[50]. We performed three rounds of library enrichment, followed by nanopore sequencing to identify the MGF proteins reacting with antibodies

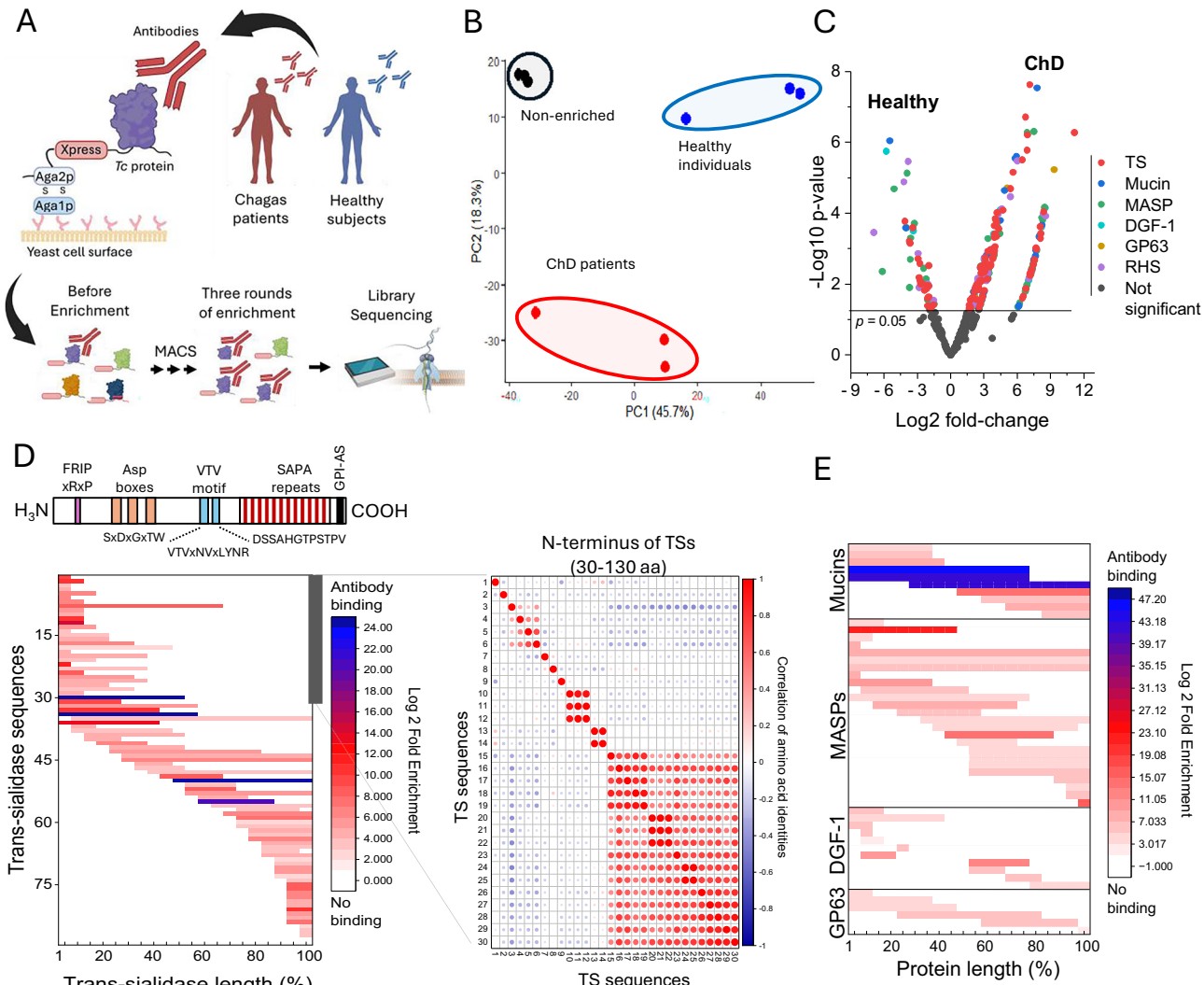

**Fig. 5 | Variable MGF sequences targeted by antibodies during human infections. A** Diagram of the YSD screen with sera from Chagas disease patients or healthy individuals. Sera was incubated with the YSD library expressing *T. cruzi* antigens on the surface and enriched using magnetic-activated cell sorting (MACS). After three rounds of enrichment, the library DNAs were sequenced using Oxford nanopore technology to identify enriched sequences compared to the initial non-enriched library or the library enriched with sera from healthy individuals. Created in BioRender. Cestari, I. (2025) https://BioRender.com/iaqrn6d. **B** PCA plot of each sequencing replicate (dots) by each enrichment condition. In red, Chagas disease (ChD) patients; blue, healthy individuals; and black, non-enriched. The experiment was performed in three biological replicates. **C** Volcano plot shows YSD screen analysis comparing MGF proteins enriched with ChD patients' or healthy individuals' antibodies. ChD or healthy enriched libraries were first compared to non-enriched libraries, and non-significant hits were removed. Only MGF proteins are shown. Statistical comparisons were performed with edgeR using a quasi-likelihood

F-test (two-sided). Effect sizes are reported as log2 fold changes and log10 *p* values. **D** Diagram of *T. cruzi* TS on top. Not all TSs have FRIP, Asp boxes, or SAPA repeats. All contain VTV motifs. Each motif's amino acid (aa) sequences are indicated, and x means any aa. The heatmap (bottom graph) displays the location of antibody binding to the TSs identified in the YSD screen, as mapped using nanopore sequencing. TS sequence lengths were normalized from 1 to 100 for visualization. On the right, sequence comparison of 30 TSs with antibodies reacting at their N-terminus (gray bar). Sequence alignment of aa 30 to 130. The first 30 aa represent predicted signal peptides removed for the analysis. The percentage of identity between all 30 sequences was obtained, and their correlation was analyzed and plotted. **E** The heat map displays the locations of antibody binding sites on mucins, MASPs, DGF-1, and GP63, which were identified in the YSD screen and mapped using nanopore sequencing. Protein sequences were normalized to a range of 1–100 for visualization. Experiments were performed in three biological replicates.

(Fig. 5A). PCA showed distinct differences in MGF proteins enriched with antibodies from patients with Chagas disease compared to healthy individuals or the non-enriched library (Fig. 5B). Comparing the screens from Chagas disease patients to healthy individuals showed sets of MGF proteins reacting specifically to Chagas disease patients' antibodies, with a significant enrichment of TS proteins (Fig. 5C, Supplementary data 2), indicating multiple MGFs expressed during human infections. The large number of TSs and MASPs is consistent with their upregulation in CTs, especially TSs (Fig. 3A-C).

We analyzed the YSD library screened with antibodies from Chagas disease patients to identify antibody binding sites within the MGF

proteins. Because we generated a fragment library, the antibody binding regions are enriched and can be identified by mining the enrichment of nanopore reads. Our *T. cruzi* genome contains 1015 TSs, encompassing all eight subfamilies and an unclassified group (Table 1). TSs are highly variable in sequence[51]. Those expressed in CTs may contain a FRIP motif, one to three Asp boxes, and a VTV motif, and some subfamilies contain the shed acute-phase antigen (SAPA) repeats at the C-terminus (Fig. 5D). Antibodies reacted to various regions in different TS sequences with no bias to a specific protein site (Fig. 5D). TSs from all eight groups reacted with human antibodies, with 37.9% of TSs being group V, 16% group II, 8% group VI, 4.6% group VIII, -2%

groups I, III, or IV, and 27.5% being unclassified TSs. The high proportion of the TS groups V and II may reflect the large number of genes within these two groups. Analysis of the N-terminus of 30 TSs (100 amino acids) reacting with antibodies showed significant divergence among the antibody targeted sequences (Fig. 5D, Supplementary Fig. 15), with half of the sequences showing lower than 20% identity (Fig. 5D, low correlation−blue dots) and the other half with up to 38% identity (Fig. 5D, moderate correlation−red dots), indicating a limited overlap in antibody binding sites. The same occurs in other parts of the protein. Mucins and MASPs were more immunogenic than DGF-1 and GP63 (Fig. 5E), consistent with their upregulation in CTs (Fig. 3A-C, Supplementary Fig. 9), with antibodies recognizing various protein sites. Analogous to TSs, their sequence diversity and variable expression suggest limited antibody cross-reaction. Hence, patient antibodies reacted to multiple MGF proteins, revealing an enrichment in TSs and implying their expression during human infections. Their sequence diversity and variable expression could help parasites evade antibody recognition during infection.

## Discussion

*T. cruzi* expanded its large gene families encoding virulence factors to ~3000 genes; however, the function and evolutionary significance of this diversification remained unknown. The families' size, repetitive sequences, and sequence similarities posed challenges in understanding their genomic organization. We generated a high-resolution genome of the *T. cruzi* Sylvio X10 strain, assembling all chromosomes and assigning precise numbers and genomic locations to MGF genes, thus complementing previous genome assemblies of this and other strains[6–8,39,40]. The assembly of most chromosomes from telomere to telomere confirmed previous observations of MGF distribution and expanded the knowledge of their genome organization. The assembly represents an advance over previous versions of the Sylvio X10/1 strain[38,39], resolving the genome into 31 chromosomes and providing a chromosome-level karyotype for a TcI Sylvio strain. While chromosome numbers might vary across DTUs, the 31 assembled chromosomes may help the analysis of other TcI strains. This improvement allows for accurate chromosomal structure and MGF distribution compared to previous assemblies. Notably, most chromosomes were diploid, except chromosomes 1 and 30, and segmental aneuploidy was found in some chromosomes, e.g., chromosome 10. Previous studies on *T. cruzi* also reported chromosome segmental aneuploidies[8]; however, low-resolution assemblies can be influenced by the resolution of repetitive regions, leading to read collapse and inaccuracies in ploidy estimations. Ploidy differences can also be due to differences among strains. Unlike other protozoan pathogens such as *T. brucei spp.* and *Plasmodium sp.*, which evolved variant antigen families in large subtelomeric arrays[3], *T. cruzi* MGF genes are typically dispersed throughout the genome in clusters, also referred to as disruptive regions[6,52], that alternate with core genes encoding housekeeping and hypothetical proteins. This is evident for TSs, mucins, MASPs, and GP63, and it may facilitate their expression, as this chromosome organization may position them near other transcribed sequences. Exceptions include the DGF-1 and RHS families, with several genes enriched in subtelomeric regions and disruptive regions in agreement with previous observations[6–8,38–40], and their genomic locations might result in differences in gene expression regulation. The size and distribution of MGFs also varied significantly across strains (Supplementary Fig. 6), also noted by others[51]. Notably, ~70% of some chromosomes− especially 29 and 31−consisted of repeat sequences containing MGF genes, suggesting their expansion in these chromosomes. The MGF gene organization differs from the earlier reports of discrete MGF clusters in scaffolds[38–40] and is well aligned with recent observations from high-resolution *T. cruzi* genome assemblies[6,8]. In contrast, most housekeeping genes were conserved and syntenic with other strains. The annotation also enabled subtype-level classification

of TSs and mucins, which helped with transcriptomic, proteomic, and antibody screening studies. The assembly provides a foundation for understanding genome architecture, antigenic diversity, and host-parasite interactions.

Our proteomic analysis revealed subsets of MGF proteins expressed at each stage of the parasite's life cycle. Moreover, MGFs were expressed from various loci across all chromosomes, as detected by RNA-seq and proteomics, indicating no specific genomic site for their expression. This is in drastic contrast with other parasites, such as *T. brucei* or *Plasmodium sp.*, which express their virulence genes from a specific locus. A low MGF gene expression level was detected by transcriptomics at the parasite population level, with some MGF transcripts undetected[52], further supporting that not all MGFs are expressed. Our TMT-MS did not detect all MGF proteins expressed in the parasite population, implying either transcriptional or post-transcriptional control of their expression. A post-transcriptional control is consistent with the 3′-UTRs of mucins and TSs regulating their expression[53,54]. The presence of chromatin three-dimensional hubs[55] could also select subsets of gene clusters for transcription and mRNAs for processing in nuclear bodies, thereby favoring the expression of different MGF genes throughout various life stages and infections. Although our proteomics data represent a population analysis, single-cell transcriptomics suggest that multiple MGF genes are transcribed in a single parasite and an upregulation of TSs in CTs, with one or a few TS transcripts expressed by each parasite[44,45]. The upregulation of TSs in CTs is also evident from our proteomic analysis and YSD screen with antibodies from Chagas disease patients. TSs function to transfer sialic acid from the host to the *T. cruzi* surface[29], helping in host cell infection and immune evasion[29]. Still, not all TS genes are functional enzymes. Based on amino acid sequence characteristics, TSs have been categorized into 8 distinct groups[50]. TS groups II, V, and VIII are the most abundant in the Sylvio X10 genome (Table 1), and all TS groups were upregulated in CTs compared to other stages. Some TSs are involved in cell invasion[28,29,32], cellular egress[21] and perhaps other functions. The mechanisms regulating TS protein (and other MGFs) expression remain unknown. We found a bimodal distribution in TS transcripts, low- and high-abundance mRNAs, suggesting a threshold-level model of expression consistent with transcript abundance regulation[53,54]. However, we cannot rule out differences at the transcriptional level. The differences in TS expression could also reflect heterogeneity in the parasite population[44], with some TSs more represented than others. In this model, the accumulation of transcripts may increase the likelihood of translation, possibly through mRNA processing and competition for the translational machinery. Hence, differences in TS mRNA production or stability could lead to their accumulation and, consequently, translation. In this context, the choice of the TS gene to be expressed is likely stochastic, resulting in a heterogeneous parasite population throughout infection.

Our data suggest a variation in MGF mRNA and protein expression in CTs. MGF expression regulation likely resets with the parasite developmental transition from AMs to CTs, contributing to different sets of MGF expressed every round of cell infection and resulting in heterogeneous parasite populations. The analysis of MGF expression after consecutive rounds of parasite infection did not reveal a specific pattern supporting programmed MGF gene expression, although we cannot rule out this possibility. The variation in MGF expression appears linked to the parasite development, e.g., from AM to CT. Variation likely also occurs after cell infection, when parasites develop from CTs to AMs, resulting in a diverse MGF repertoire expressed by AMs, as noted by proteomic analysis. Although the change in MGF expression seems linked to parasite development, other intrinsic or extrinsic factors could contribute to the observed gene expression changes. Given that only 0.2–7% of parasites are dormant during infection[56], dormancy is unlikely to be the underlying cause of

variation, but could contribute to it. *T. cruzi* infects multiple cell types, interacting with host surface receptors[28,32,33]. Diversifying surface antigens may facilitate the infection of various tissues and/or evade intracellular host cell defences. The variation may result in parasites expressing different surface proteins within a population, thereby enhancing their ability to engage with various types of host cell receptors and invade multiple tissues. Notably, multiple rounds of H9-C2 infection by CTs resulted in increased parasite fitness, i.e., increased CT production in every subsequent round of infection, suggesting regulation of genes that optimize parasite survival/proliferation in H9-C2 cells, possibly by evading intracellular host cell responses or facilitating replication.

MGF diversity and variable expression may impact host immune recognition, especially for those MGF genes encoding virulence factors such as TSs, MASPs, mucins, and GP63. The functional role of RHS and DGF-1 remains unknown. Their gene expansion, diversity, and stage-specific expression suggest potential roles in parasite survival or differentiation, intracellular host interaction, parasite–insect vector interactions, or stage-specific regulatory processes that aid parasite survival and infection. The YSD screening revealed -150 MGF antigens recognized by human antibodies. The small subset of antigens detected (-7% of the MGF repertoire) may reflect differences between the strain used to generate the YSD library and those infecting patients or the immunodominance of some MGF proteins. Alternatively, it could also reflect biases in the host immune response to some parasite proteins or differences between post-translational modifications from yeast and parasites, which could prevent antibody recognition of some yeast-expressed proteins. The limited detection of parasite MGFs is unlikely to be YSD library bias, as the library size is 30-fold of the parasite genome, with an average of 270 clones/gene, and predicted -250,000 polypeptides[48]. Interestingly, analysis of antibody binding sites revealed various immunogenic regions in different TSs, indicating the absence of a specific immunogenic region. However, some regions exhibited a high antibody signal, possibly due to the presence of multiple antibodies or a high antibody titer against particular sequences. The combination of TS sequence diversity and variation in their expression could help the parasite to escape host antibody recognition. Nevertheless, a temporal analysis of MGF gene expression variation and antibody production during infection would be necessary to establish a direct correlation. Experiments in mice or analysis of T-cells from Chagas disease patients identified different TS CD8 + T-cell epitopes[17,57,58]. The inability of CD8 + T-cells to clear *T. cruzi* infection could also result from the variable expression of TSs and perhaps other MGFs during infection. Analysis of other MGFs revealed a high antibody signal for mucins and a similar antibody binding distribution for MASPs, as observed for TSs. A potential role for the diversity and temporal changes in MGF protein expression, particularly TSs in CTs, could be to help parasites evade antibody or T-cell recognition during infection, thus contributing to parasite persistence. In this scenario, the MGF gene variation in *T. cruzi* may not be distinct from that in *T. brucei*, which is highly immunogenic and switches to evade host antibody detection[59]. *T. cruzi* MGF expression changes after host cell infection with parasites expressing genes from multiple families with a complex genome organization, and likely adapted for intracellular and extracellular parasite survival. Although the functions of many MGF genes remain unknown, their diverse repertoire may have evolved, at least in part, to facilitate host cell invasion and evade immune responses, supporting tissue tropism and parasite persistence.

## Methods

### *T. cruzi* and mammalian cell culture

EPs of TcI *Trypanosoma cruzi* strain Sylvio X10 (ATCC 50823) were cultured in Liver Infusion Tryptose (LIT) medium, supplemented with 10% fetal bovine serum and 1% penicillin-streptomycin[60]. MTs were purified from a stationary 15 mL culture of EPs grown for 12 days using

DEAE-Sephadex resin[61]. For AMs, H9-C2 cells were infected with CTs at a 1:10 ratio. The CTs were collected from the supernatant of H9-C2 (ATCC CRL-1446) after 5–6 days of *T. cruzi* infection. After 10–12 days of infection, a mix of cell culture-derived trypomastigotes and amastigotes was collected from culture supernatant, and the AM were purified using DEAE-Sephadex resin[61]. H9C2, HEK293-T, 3T3, and HeLa cells were cultured in DMEM medium supplemented with 10% fetal bovine serum (FBS) and 1% penicillin and streptomycin at 37 °C with 5% $CO_2$.

### Genome sequencing, assembly, and repeat calculation

For genomic DNA extraction, $3 \times 10^7$ mid-logarithmic growth phase parasites were centrifuged at 4000 x *g* for 10 min. The resulting pellet was washed twice with 1X PBS (37 mM NaCl, 2.7 mM KCl, 4.3 mM Na2HPO4, 1.47 mM KH2PO4), resuspended in 200 µL of lysis buffer from NEB Monarch HMW DNA Extraction (NEB #T3050, Massachusetts, USA), and processed following the manufacturer's protocol. The DNA integrity was verified by 1% agarose gel in TAE buffer (Tris 40 mM, acetic acid 20 mM, EDTA 1 mM) and quantified by Thermo Scientific™ NanoDrop™ 2000/2000c. The extracted DNA was sequenced using Pacific Bioscience (PacBio) HiFi and the Single Molecule Real-Time Technology (SMRT) Link v25.3 software at Genome Quebec, resulting in 15.8 billion bases, 1.98 million reads, with an N50 of 8976. HiFiasm v0.19.5 was used for the initial de novo assembly using HiFi PacBio sequences[62]. Pore-C reads were mapped to the assembled genome using minimap2, and the resulting BAM files were converted to BED format using the bedtools package[63]. The scaffolding was performed using Salsa with the assembled genome and Pore-C interaction data, with parameters to remove misassembled scaffolds[64] (Scripts available at https://github.com/cestari-lab/). We screened the scaffolds for telomeric repeats and evaluated repeat content to identify chromosomes. Telomeric repeats [(TTAGGG)n] were analyzed using Integrated Genome Browser (IGV, Broad Institute) motif tools and JBrowse. To assess repeats, a repeat database was built from the scaffolded genome using RepeatMasker[65]. Then, RepeatModeler was used to identify repeat families and create a de novo repeat library, followed by RepeatMasker to annotate the assembly. The masked file was used to calculate the percentage of non-masked versus masked nucleotides (the latter annotated as repeats) using a custom script (https://github.com/cestari-lab/) to determine repeat content per scaffold. Finally, the scaffolds were sorted and aligned based on the *T. cruzi* Brazil A4 and Dm25 TcI reference strains[7,8] using the AssemblyReferenceSorter tool in the Next Generation Sequencing Experience Platform (NGSEP) v4.3.1[66], resulting in the final set of chromosomes and scaffolds. Synteny analysis was performed to compare the *T. cruzi* Sylvio genome with other TcI strains, including Brazil A4 v.68 (downloaded from TriTrypDB repository), Dm25 (downloaded from NCBI), CL-Brenner Esmeraldo like v.68 (downloaded from TriTrypDB repository), and the previous version of Sylvio X10 v.68[67]. Genomic alignments were generated using minimap2, producing PAF files visualized in JBrowse2 or plotted in R software using the Circlize package to assess structural rearrangements and conserved regions. Genome-wide Average Nucleotide Identity (ANI) was calculated with FastANI to quantify the similarity between genomes. Chromosome-level and scaffold-level were also compared using the same approach.

### Chromatin conformation capture and nanopore sequencing (Pore-C)

*T. cruzi* Sylvio X10 EPs at mid-logarithmic growth ($1.8 \times 10^9$ parasites total) were fixed in 1% paraformaldehyde for 10 min, rotating at 150 rpm, then quenched with 0.2 M glycine. Cells were centrifuged at $4000 \times g$ and washed three times in PBS. Pellets were resuspended in 1 mL hic-lysis buffer (10 mM Tri-HCl pH 8, 10 M NaCl, 1X Roche protease inhibitor cocktail) and rotated for 30 min at 4 °C. They were then pelleted at $2500 \times g$ for 5 min at 4 °C. The pellet was then resuspended

in 0.5% SDS at 62 °C for 10 min, followed by adding 1.5% Triton X-100 and incubating at 37 °C for 15 min. The sample was digested with 100 units of Hind III HF (New England Biolabs) at 37 °C overnight. Samples were incubated for 20 min for heat inactivation. DNA ends were filled in with 10 mM biotin-dCTP (Jena Bioscience) and non-biotinylated 10 mM dATP, dGTP, and dTTP with 50 units of DNA polymerase I, large Klenow fragment (New England Biolabs) at 16 °C at 80 rpm overnight. DNAs were extracted with phenol:chloroform:isoamyl-alcohol (25:24:1, pH 6.7) and precipitated with 100% cold isopropanol for 30 min at $14,000 \times g$ at 4 °C. The pellet was washed with 75% ethanol and resuspended in 10 mM Tris pH 8. Streptavidin magnetic beads (New England Biolabs) were added and rotated at 150 rpm for 45 min at room temperature (RT), then washed three times in 10 mM Tris pH 8 at 55 °C for 2 min. DNAs were separated using a magnetic rack, eluted in RNA-free water, and incubated with 40 µg of Proteinase K at 55 °C overnight and shaking at 150 rpm. DNAs were separated on a magnetic rack and then extracted with phenol:chloroform:isoamyl alcohol (25:24:1, pH 6.7), as above. Eluted DNAs in RNase-free water were used for Oxford nanopore library preparation and DNA sequencing, as previously described[68]. The output.fastq files were mapped to the genome using minimap2[69] and DNA contacts were analyzed using pairtools[70]. The scripts used for analysis are available at https://github.com/cestari-lab.

## Chromosomal numbers determination by depth

The *T. cruzi* Sylvio X10 strain assembled genome (this work) was used as an index for minimap2 alignment with PacBio HiFi reads[69]. For ploidy determination, the standardized protocol by Schwabl et al. was followed[71]. The Samtools depth tool (v0.1.18) was used to calculate the average read depth for 1 kb bins across each chromosome from the.bam files generated by minimap2 using PacBio or Oxford nanopore sequences. The median of the average depths was then calculated for all bins. Somy was estimated by dividing the average depth by the median at the 60th percentile and multiplying the result by two.

## Genome annotation

The *T. cruzi* Sylvio X10 genome annotation was performed using the GenSAS platform (Supplementary Fig. 16). Annotation included EST data from *T. cruzi* sequencing libraries available in TrytripDB (https://tritrypdb.org/), companion gene predictions based on the CL-Brenner reference, amino acid FASTA file containing all available *T. cruzi* proteins from TrytripDB, nucleotide FASTA for multigene family proteins from TrytripDB, a GFF file of repeats obtained from RepeatMasker, and paired-end RNA-seq reads from trypomastigotes (sequencing read archive identification: SRR9202394). The reads were mapped to the assembly using HISAT v2.2.1. Initial alignments against the NCBI RefSeq Protozoa nucleotide database were performed using BLASTn v2.12 with specific parameters (Expect value 1e-50, 85% identity, maximum of 10 hits per region, Gap Open of 5, and Gap Extend of 2), along with BLAT v2.5 with a minimum identity of 95%. Alignments were also conducted against the EST evidence and multigene family FASTA files using the same BLASTn and BLAT parameters. PASA v2.4.1 was used to perform spliced alignments of expressed transcript sequences to aid in modeling gene structures. For structural annotation, we used Augustus v3.4.0 for gene prediction, incorporating data from NCBI RefSeq Protozoa and multigene families, supported by gene structures generated through PASA, and protein databases, and RNAseq alignments. Additional ab initio gene prediction was performed using GeneMarkES v4.48, and gene identification was further refined using SNAP v2017-05-17. The results were integrated with EvidenceModeler, which combines ab initio predictions with protein and transcript alignments into a consensus gene structure and refined using PASA. For functional annotation, BLASTp was employed with LOSUM62 matrix, Expect value 1e-8, Word Size 3, Gap Open 11, Gap Extend 1, and a maximum HSP distance of 30,000. DIAMOND v2.0.11 was used to compare against the NCBI RefSeq Protozoa protein dataset and the protein sequences from TrytripDB. InterProScan was run to identify protein signatures from the InterPro consortium, while the Pfam database was used to assign protein families. SignalP predicted the presence and location of signal peptides, and TargetP predicted subcellular localization for eukaryotic proteins. The outputs from the tools were manually inspected and merged to produce the final annotated genome. To annotate subgroups of TSs, annotated TSs were blasted against the TS subgroups from Dm28c genome.

## Proteomic sample preparation

For EPs, a fresh 80 mL culture at a concentration of $1 \times 10^6$ parasites/mL was maintained for 72 h (logarithmic phase), and the absence of MTs was verified under a microscope. The culture was centrifuged at 4000 g for 5 min. Parasites were washed twice with 1X PBS, transferred to Eppendorf tubes, centrifuged once at 4000 g for 5 min, and the pellet (EPs) was collected. MTs, AMs, and CTs were purified as described above (see *T. cruzi and mammalian cell culture*), washed twice in 1X PBS, and pellets were collected. Parasite pellets were snap-frozen in liquid nitrogen and stored at −80 °C until protein extraction. Protein extraction was performed by resuspending cell pellets in lysis buffer (60 mM Triethylammonium bicarbonate buffer (TAEB), 8 M urea, 1 mM Ethylenediaminetetraacetic acid (EDTA), and 2x protease inhibitors). Samples were sonicated in Covaris M220 ultra sonicator (75 peak power, 10% duty cycle, 200 cycles, and a 100-s duration). Afterward, lysates were transferred to Eppendorf tubes and centrifuged at $10,000 \times g$ for 10 min. Extracted proteins (supernatants) were collected and stored at −80 °C. Aliquots were collected for SDS-PAGE and proteins quantified using a BCA kit (Life Technologies) according to the manufacturer's instructions. Four biological replicates were generated for each parasite stage.

## TMT-labeling and mass spectrometry

A 100 µg of extracted parasite proteins (EP, MT, AM, and CT) was adjusted to 100 µL with 100 mM TEAB. Samples were reduced with 5 mM TCEP at 55 °C for 1 h and alkylated with 18.75 mM iodoacetamide for 30 min at room temperature in the dark. Proteins were precipitated overnight with six volumes of pre-chilled acetone at −20 °C, pelleted by centrifugation, and resuspended in 100 mM TEAB pH 8.5. Digestion was performed overnight at 37 °C using trypsin at a 1:40 (w/w) enzyme-to-protein ratio. The peptides were treated with TMT-16plex reagents (ThermoFisher Scientific) according to the manufacturer's instructions. The labeled peptides were fractionated using the Pierce™ High pH Reversed-Phase Peptide Fractionation Kit into 8 fractions. Each fraction was re-solubilized in 0.1% aqueous formic acid, and 2 micrograms of each fraction were loaded onto a Thermo Acclaim Pepmap precolumn (75 µm ID × 2 cm, C18, 3 µm beads) and then onto an Acclaim Pepmap Easyspray analytical column (75 µm x 15 cm, 2 µm C18 beads) for separation using a Dionex Ultimate 3000 uHPLC system at a flow rate of 250 nL/min. A gradient of 2–35% organic solvent (0.1% formic acid in acetonitrile) was applied over three hours, using default settings for MS3-level SPS TMT quantitation[72] on an Orbitrap Fusion mass spectrometer (ThermoFisher Scientific) operating in DDA-MS3 mode. Briefly, MS1 scans were acquired at a resolution of 120,000, scanning from 375–1500 m/z, with ions collected for 50 ms or until an AGC target of 4e5 was reached. Precursors with a charge state of 2–5 were selected for MS2 analysis, using an isolation window of 0.7 m/z. Ions were collected for up to 50 ms or until an AGC target of 1e4 was reached, and fragmented using CID at 35% energy. MS2 spectra were read out in the linear ion trap in rapid mode. From the MS2 spectra, the top 10 precursor notches (based on signal height) were selected for MS3 quantitative TMT reporter ion analysis. These were isolated with a 2 m/z window, fragmented with HCD at 65% energy, and the resulting fragments were read in the Orbitrap at a resolution of 60,000, with a maximum injection time of 105 ms or until an AGC target of 1e5 was

reached. Raw.raw files were processed using Proteome Discoverer 2.2 (ThermoFisher Scientific) to identify proteins and quantify TMT reporter ion intensities. Standard TMT quantification workflows were used, with Trypsin set as the enzyme specificity. Spectra were matched against a strain-specific *T. cruzi* database. Dynamic modifications were set as oxidation of methionine (M) and acetylation at protein N-termini. Cysteine carbamidomethylation and TMT tags at both peptide N-termini and lysine residues were set as static modifications. The results were filtered to a 1% FDR to ensure high confidence in protein identification based on ANOVA with a 95% confidence level for individual proteins and *p*-value adjusted using the Benjamini-Hochberg method.

### Universal MGF primer design

To evaluate expression changes in MGFs, we designed universal primers to amplify all genes from a gene family. Available sequences of MGFs (Mucins, MASPs, TSs, RHS proteins, DGF-1, and GP63) were obtained from strains Dm28c (2018), Brazil A4, Y C6, CL Brener Esmeraldo-like, CL Brener non-Esmeraldo-like, available in the Tri-TrypDB database, and Sylvio X10 assembly genome (this work). For each group of MGFs, fasta sequences were aligned using MAFFT, and primers were designed based on conserved regions located at the 3' conserved ends of the sequences. Due to the high genetic variability within some MGFs, multiple primers were developed to ensure comprehensive coverage. For TSs, primers were designed for each subgroup (I-VIII) using sequences from Dm28c and Sylvio X10 genome (this work) and assessed for compatibility with all annotated TSs in the other genomes. Primer designs were evaluated for secondary structure formation, specificity, and melting temperature (TM) compatibility with *T. cruzi* spliced leader sequences. The final set of primers (Supplementary Table 2) was tested using DNA from *T. cruzi* EPs to confirm amplification efficiency and reliability.

### MGF-Seq

MTs were used to infect H9-C2 cells (ratio 1:10) and incubated in DMEM medium supplemented with 10% FBS and 1% penicillin and streptomycin at 37 °C with 5% $CO_2$. After 10 days, the CTs produced were collected and quantified to re-infect H9-C2 cells, and the process was repeated to produce a total of four consecutive CT generations. After the third generation, CTs were used to infect HEK293T cells. After six days CTs were collected. The procedure was repeated to obtain four biological replicates. MTs and CTs were washed twice in 1X PBS, and RNAs were extracted using the RNeasy® Plus Mini Kit, which removes DNA. cDNAs were synthesized using the LunaScript® RT SuperMix Kit (NEB) according to the manufacturer's instructions. cDNAs were diluted 10X in water and MGF genes amplified using Taq DNA Polymerase with ThermoPol® Buffer (NEB), according to manufacturer's protocol, using a splice leader primer and universal 3' primers for MGF genes (Supplementary Table 2) with an initial denaturation of 95 °C for 5 min, and 22 cycles of 95 °C for 35 s, 59 °C for 45 s, and 68 °C for 2 min, and final extension of 68 °C for 5 min[46,47]. PCR products were purified using the NucleoMag® NGS Clean-up and Size Select kit (Takara), retaining amplicons larger than 500 bp. DNA fragments were prepared for Oxford Nanopore sequencing using the Ligation Sequencing Kit (SQK-LSK114, Oxford Nanopore Technologies) and the PCR Barcoding Expansion Kit (EXP-PBC096) following the manufacturer's instructions, with cDNA from biological replicates and infection generations receiving different barcode sequences (26 samples total) and multiplexed. Fifty femtomoles (fmol) of pooled and barcoded libraries were loaded onto a MinION sequencing device with an R10 flow cell (FLO-MIN114, Oxford Nanopore Technologies) and sequenced for 72 h using MinKNOW 1.4.2 (Oxford Nanopore Technologies) software, yielding ~15 Gb of data. Total RNA-seq data of CTs were obtained from the Sequence Read Archive (SRA) accession SRR9202394, SRR2177699 and SRR2177698.

### Quantifying *T. cruzi* invasion and infection

A total of 30,000 H9-C2, HEK, HeLa, and 3T3 cells were incubated at 37 °C with 5% $CO_2$ in RPMI medium supplemented with 10% FBS and 1% penicillin and streptomycin for 4 h in 24-well plates to allow adherence. Cells were washed twice with 1X PBS and subsequently infected with Sylvio X10 *T. cruzi* MT or CT at a parasite-to-host cell ratio of 1:10. After 3 h of incubation, the cells were washed to remove unbound parasites from the supernatant and fixed using 4% formaldehyde. Cells were mounted using Fluoromount with DAPI. Stained DNAs of mammalian cells and parasites were used to quantify infections using a Citation 5 Imaging System under a 40x objective using Gen5 software 3.17.17 (BioTek). Three biological replicates were included. The data were analyzed using a t-test for pairwise comparisons with a 95% confidence level.

### YSD screen using Chagas disease patient antibodies

A genome-wide yeast surface display was constructed for *T. cruzi* Sylvio X10 and transformed into EBY100 *Saccharomyces cerevisiae*[48,68]. To express *T. cruzi* proteins on the yeast surface, $1.2 \times 10^7$ yeast cells were grown in 5 ml YPD media at 30 °C and 225 rpm until OD600 1 (~2 h). Cells were collected by centrifugation at 3000 x g for 5 min and washed three times in 10 ml sterile MilliQ water by centrifugation at 3000 x g for 5 min. Cells were resuspended in 50 ml SD/-trp media with 2% dextrose (SD/-trp+dex) and incubated at 30 °C and 225 rpm until OD600 reached 1 (~16 h). Cells were collected by centrifugation and washed in water, as indicated above. Cells were resuspended in 50 ml SD/-trp with 2% raffinose and incubated at 30 °C and 225 rpm for 2 h. Cells were collected by centrifugation, resuspended in 50 ml SD/-trp + 2% raffinose + 1% galactose for protein expression, and incubated at 30 °C and 225 rpm for 16 h. For binding assays, yeasts were collected by centrifugation and washed three times in ice-cold PBS (as indicated above) and incubated in 1:1000 pooled sera from five chronic-stage Chagas disease patients diluted in PBS or 1:1000 pooled sera of two healthy individuals. Sera was kindly provided by Dr. Momar Ndao (McGill University Health Centre) from patients with Chagas disease or healthy individuals, who were Latin American migrants from Bolivia, with an age range of 28–56 years[73]. Written informed consent was obtained from all participants[73]. Serological tests were used to confirm the status of patients and control sera[73]. Yeast and sera mix were incubated, rotating for 2 h at 4 °C. Cells were collected by centrifugation and washed three times in PBS. Cells were resuspended in 400 μL of 10 mM PBS and 100 μL magnetic Protein G beads and incubated at 4 °C rotating for 30 min. The 500 μL yeast and magnetic bead mixture was loaded onto a Miltenyi Biotech column and passed through by gravity on a magnetic stand. The bound yeasts were washed in the column three times in PBS to remove unspecific binders. Antibody-bound yeasts were removed from the magnetic rack and re-cultured to expand the enriched (binders) population in SD/-trp + 1% dextrose overnight. This process was repeated three times for both conditions to enrich antibody-binding populations. To identify antibody-binding proteins, the yeast cells from expanded populations were collected and washed three times to remove media and plasmid DNAs were extracted using an adapted protocol from[74], where DNA is purified using NGS magnetic beads at 0.7x ratio rather than isopropanol precipitation. Oxford nanopore library and sequencing were prepared as previously described[75]. Scripts used for computational analysis can be found at https://github.com/cestari-lab/YSD. Briefly, Oxford nanopore sequence.fasta files were aligned to the genome using minimap2. Sequence Alignment Map (.sam) files were converted to Binary Alignment Map (.bam) files using the samtools package, filtering supplementary and secondary reads and any reads with a MAPQ score <1. The coverage was analyzed using plotCoverage from deeptools. The resulting.bed file was visualized using Integrative Genomic Visualizer (IGV, Broad Institute). Read counts were obtained using feature-Counts, and fold change comparisons between groups were

performed using edgeR and enriched regions with macs3. Heatmaps were generated from edgeR enrichment analysis.

**Reporting summary**

Further information on research design is available in the Nature Portfolio Reporting Summary linked to this article.

## Data availability

The assembled genome of the *T. cruzi* Sylvio X10 strain is available in the National Center for Biotechnology Information (NCBI) with Bio-Project number PRJNA1237338 and at the DNA DataBank of Japan (DDBJ), the European Nucleotide Archive (ENA), and GenBank at NCBI under the accession JBMETK000000000. PacBio HiFi sequences and Pore-C data are available in the Sequence Read Archive (SRA) with the BioProject identification PRJNA1236874 [https://www.ncbi.nlm.nih.gov/bioproject/ PRJNA1236874/]. The mass spectrometry proteomics data have been deposited to the ProteomeXchange Consortium via the PRIDE partner repository with the dataset identifier PXD061891. Source data are provided with this paper.

## Code availability

Codes used for data analysis are available at https://github.com/cestari-lab/. For coding citation, please refer to MGFseq (10.5281/zenodo.17137432), genome assembly (10.5281/zenodo.17137452), and other lab codes (10.5281/zenodo.17137473).

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

## Acknowledgements

Canadian Institutes of Health Research grant CIHR PJT-175222 (IC). The Natural Sciences and Engineering Research Council of Canada (NSERC) grant RGPIN-2019-05271 (IC). Canada Foundation for Innovation grant JELF 258389 (IC). CIHR-IDRC-ISF Joint Canada-Israel Health Research Program Phase II grant IDRC 109929 (IC). William Dawson Scholar Award 101157 (IC). NSERC CGS M fellowship (LBA). NSERC CGS D fellowship (ML). FRQNT PBEEE postdoctoral training scholarship 351627 (LCS). This research was partly enabled by computational resources provided by Calcul Quebec (https://www.calculquebec.ca/en/) and the Digital Research Alliance of Canada (alliancecan.ca).

## Author contributions

LBA and ML extracted *T. cruzi* DNA for genome sequencing. LCS designed a genome assembly pipeline and performed computational analysis. LCS performed *T. cruzi* infections, purifications, and sample preparation for mass spectrometry and invasion assays. LCS analyzed mass spectrometry data. IC and LCS designed MGF-seq, and LCS performed MGF-seq and computational analysis. ML and LBA transformed *T. cruzi* genome-wide in yeast, and ML performed the YSD screen and computational analysis. IC designed the research, analyzed the results, secured research funding, supervised the study, and wrote the manuscript. IC and LCS reviewed and edited the manuscript. All authors read and revised the manuscript.

## Competing interests

The authors declare no competing interests.
