## [Transparent Peer Review file · Nature Communications]

Variation in surface protein expression leads to heterogeneous *Trypanosoma cruzi* populations during host cell infection

Corresponding Author: Professor Igor Cestari

Version 0:

Reviewer comments:

Reviewer #1

(Remarks to the Author)

This manuscript presents a chromosome-level assembly of the *T. cruzi* Sylvio X10 strain (TcI) genome, utilizing PacBio HiFi long reads integrated with chromatin conformation capture via Nanopore-based Pore-C. This approach results in a 38.1 Mb assembly with an N50 of 1.3 Mb and 31 assembled chromosomes (27 of which are complete end-to-end). Achieving telomere-to-telomere contiguity for most chromosomes represents a significant technical strength, given the challenges associated with assembling *T. cruzi* multigene families. The authors further validate the accuracy of the assembly through depth-based ploidy analysis, revealing that most chromosomes are diploid with a few aneuploidies (one haploid and one triploid chromosome). Overall, the genome assembly approach is rigorous and provides a solid foundation.

Next, the authors combine TMT-based quantitative proteomics with RNA-seq to assess gene expression across the life stages, identifying hundreds of differentially expressed proteins. As expected, trans-sialidases, mucins, and MASP are upregulated in cell-derived trypomastigotes. In contrast, the DGF-1 and RHS families are enriched in epimastigotes or amastigotes. While the proteomic resolution for highly similar gene family members is limited, integration with RNA-seq and a targeted nanopore-based "MGF-seq" confirmed the broad expression of MGF genes. A bimodal distribution of TS transcripts suggests post-transcriptional regulation, though detection limits in proteomics may still be a concern. Overall, the methods are robust and innovative, particularly in the temporal tracking of gene expression over serial infections. Finally, the Yeast Surface Display (YSD) Immunoscreen is a highlight of the strategy, showcasing creativity in addressing the immunological relevance of the parasite's gene diversity.

This manuscript presents relevant and original findings by combining cutting-edge genomic, proteomic, and immunoscreen approaches. However, several aspects of the study need to be revised, as detailed below.

Major comments

1- Overall, the manuscript presents a substantial amount of data. While the authors provide numerous observations and analyses, the work would benefit significantly from a more precise articulation of the overarching scientific question or hypothesis being addressed. Currently, the central aim of the study remains implicit. It is essential for the authors to clearly state the main research question early in the manuscript and ensure that each section is structured to contribute directly to answering that question. Furthermore, each section should be more tightly integrated into the narrative, with improved transitions and logical flow.

2- A major concern is that, throughout the manuscript, the authors seem to follow a narrative based on the "pre-long-read sequencing/Hi-C era," relying on concepts that date back to the publication of the first draft genome of *T. cruzi* in 2005. At that time, the genome's highly repetitive nature, combined with the limitations of Sanger sequencing, resulted in a highly fragmented assembly. This fragmentation led to the concept of "multigene family (MGF) clusters," as it was not feasible to infer higher-order genome organization. However, since the first long-read-based *T. cruzi* genome assemblies, it has become increasingly clear that the genome is much more structured than previously thought. A conserved core region has been identified, showing synteny with *T. brucei* and *Leishmania*. In contrast, a disruptive region—characteristic of *T. cruzi*—is enriched in large arrays of mucins, MASPs, and trans-sialidases (TS). These disruptive regions are GC-rich and span

extensive, continuous tracts of the chromosomes. In fact, entire scaffolds can now be classified as mainly core, mainly disruptive, or mixed, whereas in the current manuscript, the authors refer to the presence of MGF tracts as if this were a novel observation. This interpretation needs substantial revision, considering the existing body of work (see references at the end of this point).

Additionally, these compartments vary in their nucleosomal organization and chromatin structure, contributing to their spatial segregation into distinct nuclear territories—specifically, 3D compartments C and D. Moreover, the DGF-1 and RHS families have consistently been associated with subtelomeric regions (the first publications are at least from 2005), suggesting they may play unique functional roles. These gene families should be discussed separately rather than treated as part of a general multigene family group. Finally, the trans-sialidase gene family comprises at least eight groups of TS genes that have been described. One of these groups encodes catalytically active enzymes with bona fide TS activity, while the remaining groups are functionally diverse. This functional and structural classification must be clearly considered when discussing the TS family and their patterns of expression.

As just one example, in the abstract, the sentence “The genome revealed multigene families of virulence genes accounting for ~70% of some chromosomes, organized in 15 clusters or scattered through housekeeping genes” seems to present previously established findings as if they were novel.

Several other statements throughout the manuscript describe well-known observations without citing prior literature, which may inadvertently convey a sense of novelty for findings that have already been reported. For instance:

Line 98 – “In contrast, DGF-1 and RHS were enriched in subtelomeric regions.” → This has been previously established.

Lines 99–100 – “MGF sequences correlated with a high GC content and were rich in LINE/I-L1Tc retrotransposons and RHS.” → Not all MGFs are GC-rich, and RHS genes, being mainly subtelomeric, should not be grouped indiscriminately with the rest of the MGFs.

Line 122 – “We identified a stage-specific pattern of MGF protein expression, characterized by variations in protein abundance across each stage. TSs, mucins, and MASPs were significantly upregulated in CTs.” Stage-specific expression of these multigene families has already been reported.

Line 125 – “whereas GP63 was up-regulated in AMs and CTs.” → Also previously described.

A particularly important example appears in the discussion (lines 210–215):

“*T. cruzi* MGF genes are typically dispersed throughout the genome in clusters that alternate with housekeeping genes. This is evident for TSs, mucins, MASPs, and GP63, and it may facilitate their expression, as this chromosome organization may position them near other transcribed sequences. Exceptions include the DGF-1 and RHS families, which are enriched in subtelomeric regions. Notably, ~70% of some chromosomes consisted of repeat sequences containing MGF genes, suggesting their expansion in these chromosomes.”

This entire passage recapitulates well-established observations without citing any previous work. As noted above, these features have been known for some time, and the lack of attribution is a serious oversight that should be addressed.

3. Misuse of the Term “Stochastic” and Need for Reframing the Interpretation

The use of the term stochastic in the manuscript is potentially misleading. The observed variability in gene expression reflects an underlying heterogeneous parasite population, as demonstrated by single-cell RNA-seq studies. These studies (<https://doi.org/10.1101/2024.10.01.616042>; doi: <https://doi.org/10.1101/2025.01.14.633000>) have shown that multigene family expression varies significantly from one parasite to another.

This has important implications, requiring reconsideration of both the title and the discussion. Rather than framing the observed variability as stochastic, the authors should emphasize that expression heterogeneity in *T. cruzi* is a well-established phenomenon. What should be highlighted as a novel contribution of the present study is the observation that passage through cultured host cells already imposes selective pressure, leading to shifts in subpopulation composition. This suggests that multigene family expression can be modulated by intrinsic factors or environmental cues, even in the absence of immune selection.

At that point in the discussion, the authors may suggest a stochastic element as a contributing factor, but it should not be presented as the central mechanism. For example, if this process were similar to VSG switching in *T. brucei*, where clonal variation is followed by immune-driven selection, one would not expect passage through cultured host cells to be sufficient to shift expression patterns. The fact that it does suggests a high capacity for intrinsic plasticity in *T. cruzi*.

Importantly, dormancy could be the underlying cause of the observed shifts. A subpopulation may remain in a transcriptionally silent (or low-activity) state in a stochastic manner, and upon successive passages, different subpopulations may become dominant. This possibility provides a plausible mechanistic explanation and should be discussed more explicitly in the manuscript.

3- Concerning ploidy:

I strongly suggest mapping a deep set of Illumina reads to increase the sensitivity of the analysis. In addition, the distinction between chromosomes 22 and 22b remains unclear. Chromosome 22 is described as monosomic, yet the figure also shows a chromosome labeled 22b, which appears to display a monosomic pattern as well. Clarification of the relationship between these two chromosomes is needed.

Lines 83-84: "Sequence depth confirmed chromosome copy numbers and indicated segmental aneuploidy in chromosome 1." It is unclear what the authors mean by "segmental aneuploidy." Please revise the figures.

4. The authors report transcriptomic differences across *T. cruzi* life cycle stages based on an MGF-enriched transcriptome. However, this approach does not eliminate the possibility of technical bias introduced by the enrichment process. It is unclear why a standard bulk RNA-seq was not performed in parallel to provide an unbiased, genome-wide expression profile. We strongly recommend that the authors compare the differential expression patterns they report at least with those obtained from other publicly available RNA-seq datasets, such as the bulk RNA-seq used to validate the single-cell RNA-seq atlas [doi: <https://doi.org/10.1101/2024.10.01.616042>]. This comparison would help validate the observed expression trends and determine whether they are consistent with prior observations or possibly skewed by the targeted approach. Clarifying this point is crucial, especially when interpreting gene family expression dynamics that are known to be highly variable and sensitive to both biological and technical factors.

Minor comments.

1- While this deep sequencing approach is robust, it is necessary to consider whether highly repetitive gene family regions still pose assembly ambiguities. Indeed, the authors find 39 short unplaced scaffolds (6–68 kb) that likely represent segmental aneuploidy or divergent haplotype segments with multigene family duplications. These residual scaffolds suggest that some repeats or haplotype-specific sequences could not be perfectly resolved. I ask the authors to clarify if these unincorporated segments contain additional gene copies or allelic variants of virulence genes.

2- Regarding the above, it is essential to clarify whether multimapping reads still exist (as with short read sequencing) and how to handle them, or whether PaBioHiFi is sufficient to clearly distinguish highly homologous sequences coding for MGF.

3- The phrase "...infection, suggesting epigenetic regulation..." is not supported by data and should be removed or rephrased to reflect its speculative nature.

4-166: "The increased production of CTs after consecutive H9-C2 infections (Fig. 4D) may be due to increased AM proliferation fitness rather than selection of CTs for invasion." This interpretation could be directly tested by quantifying intracellular amastigotes. The phrase "may be due to" should be avoided in the absence of supporting data.

5- In the circos diagram, there is no clear rationale for grouping MASP, mucins, and GP63 together while treating TS separately. The grouping should be biologically justified or revised for consistency.

6- Several figures present schematic representations rather than actual results. For instance, the pipeline diagram (Fig. 1A) and the protein labeling schematic (Fig. 2B) do not convey original data and could be moved to supplementary material or replaced with concise textual descriptions.

7. These figures should be reformulated to reflect the distinction between core and disruptive compartments, which is central to current genome organization models in *T. cruzi*.

(Remarks on code availability)

The code availability allows to download and analyze all of the data.

Reviewer #2

(Remarks to the Author)

In this manuscript, Saavedra and colleagues seek to deepen understanding of the functions provided by the very extensive multigene families (MGFs) in the genome of *Trypanosoma cruzi*. The findings of the manuscript can be summarised as follows: a new genome assembly of *T. cruzi* Sylvio X10 strain, clarifying the number of chromosomes and level of aneuploidy; demonstration of life cycle changes in the expression of several MGFs at both the protein and RNA level; and demonstration of immune recognition of several MGF proteins in Chagas patients. These are valuable insights, where the improved genome assembly has allowed much greater clarity of MGF expression dynamics across the *T. cruzi* life cycle, and has allowed defined understanding of immune recognition of the MGFs. If communicated carefully, the manuscript will be of value, but I'm afraid the title, abstract and results are worded in ways that exaggerate the findings, and should be changed, as explained below:

There are two problems with the presentation of the work:

1. The proteomics and RNAseq experiments rely on population-levels analysis, which mainly shows that different MGFs show maximal expression in different life cycle forms, and expression of these MGFs appears to be just a portion of the overall repertoire. Thus, it is inaccurate to say that there is 'stochastic variation in surface protein expression' (eg title), since the available data can be interpreted as programmed changes in MGF expression; have the authors asked if the limited expression of some MGFs (eg TSs in Fig.4) is due to expression of functional genes and non-expression of pseudogenes? Can they show changes in patterns of MGF parts of the families that might provide evidence of stochastic or programmed expression change? It is also not at all reasonable in the results (eg lines 130-131 and 153-156) to invoke comparisons with single-cell RNAseq data; this study cannot, in any way, determine if there is cell-to-cell MGF expression heterogeneity.

(Please also note that the Inchausti et al paper (ref 33) was published in 2025, not 2005.)

2. The authors try to suggest that their data demonstrates MGF expression variation allows 'the invasion to multiple tissues'(abstract), and 'help[s] parasites avert antibody recognition' (abstract). Both conclusions significantly overstate the findings: the authors merely measure infection levels in two cultured host cell types, and demonstrate antibody responses to MGF proteins (but did not correlate changes in MGF expression with timing of antibody responses). These are reasonable speculations if limited to the discussion, but require significantly greater evidence to be presented as main conclusions, as they currently are (eg 'diversifies T. cruzi infection'; title).

More specific comments:

3. Line 45. Provide references for parasite evasion of immunity in certain tissues.

4. Introduction. For a non-specialist reader, it would aid clarity to outline the T. cruzi life cycle forms and where they are found.

5. The description of the new genome assembly is very abbreviated and warrants some expansion. What are the implications of now defining the genome as comprising 32 chromosomes: what was known before; is this likely to be specific to this strain; can they summarise what was learned by comparing this genome assembly with several other strains (Fig.S4)?

6. Is the pore-C data shown anywhere, and how does this compare with previous Hi-C work from the Robello lab?

7. Line 76. In what way can Nanopore sequence 'validate' the PacBio assembly?

8. Line 80. Please clarify why chromosomes 22 and 22b are considered different chromosomes and not merely alleles of a diploid chromosome (is a large insertion enough to make them distinct chromosomes?).

9. Line 84, Fig.1C. Can the authors clarify how these data reveal segmental aneuploidy; can these contigs not represent extrachromosomal elements? Please also discuss how the ploidy/aneuploidy detected here compares with previous literature.

10. Line 130 is a good example of over-interpreting the data: 'RNA-seq and proteomic analysis confirmed the expression of large subsets of MGF genes from multiple chromosome loci (Fig. 3E, Fig. S8). This agrees with single-cell data suggesting transcription of TSs from multiple loci in different cells (33). The diversity of MGF proteins expressed likely reflects heterogeneity within the parasite population.' There is no comparison made between these data and scRNAseq, and all these data can be interpreted as programmed expression of a selection of MGFs across the cell cycle.

(Remarks on code availability)

Reviewer #3

(Remarks to the Author)

Stochastic variation in surface protein expression diversifies *Trypanosoma cruzi* infection

In this study by Saavedra et al, the authors describe variation in multigene families in *T. cruzi* and how this may aid parasite persistence in the host. By integrating high-resolution genome assembly, proteomics, and genome-wide yeast surface display, they argue that variation in the expression of multigene families facilitates host cell invasion and to evade hosts immune response and how stochastic variation in the expression of these MGF contributes to the diversification of parasite populations. Overall, this is a well-executed and insightful study that advances our understanding of how *T. cruzi* leverages gene family variability to sustain chronic infection.

Main points:

1. The inclusion of a short paragraph in the introduction on the *T. cruzi* life cycle and development / expression of MGF would benefit the non-specialist reader.
2. There are several studies that report high-resolution sequencing and genome assembly of *T. cruzi* strains, including Sylvio X10, yet not referenced. Included in the discussion there should be a comparison of the assembly done here with published studies.

These include:

- Wang W, Peng D, Baptista RP, Li Y, Kissinger JC, Tarleton RL (2021) Strain-specific genome evolution in *Trypanosoma cruzi*, the agent of Chagas disease. *PLoS Pathog* 17(1): e1009254. <https://doi.org/10.1371/journal.ppat.1009254>
- Callejas-Hernández, F., Rastrojo, A., Poveda, C. et al. Genomic assemblies of newly sequenced *Trypanosoma cruzi* strains reveal new genomic expansion and greater complexity. *Sci Rep* 8, 14631 (2018). <https://doi.org/10.1038/s41598-018-32877-2>
- Franzén O, Ochaya S, Sherwood E, Lewis MD, Llewellyn MS, Miles MA, Andersson B. Shotgun sequencing analysis of *Trypanosoma cruzi* I Sylvio X10/1 and comparison with *T. cruzi* VI CL Brener. *PLoS Negl Trop Dis*. 2011 Mar 8;5(3):e984. doi: 10.1371/journal.pntd.0000984. PMID: 21408126; PMCID: PMC3050914

- Díaz-Viraqué F, Chiribao ML, Libisch MG, Robello C. Genome-wide chromatin interaction map for *Trypanosoma cruzi*. *Nat Microbiol.* 2023 Nov;8(11):2103-2114. doi: 10.1038/s41564-023-01483-y. Epub 2023 Oct 12. PMID: 37828247; PMCID: PMC10627812.
- Dean AAC, Berná L, Robello C, Buscaglia CA, Balouz V. An algorithm for annotation and classification of *T. cruzi* MASP sequences: towards a better understanding of the parasite genetic variability. *BMC Genomics.* 2025 Feb 24;26(1):194. doi: 10.1186/s12864-025-11384-5. PMID: 39994548; PMCID: PMC11852901.

3. The MGF-seq is good idea, however it follows the same principle as VSG-seq from Mugnier MR, et al. *Science.* 2015 PMID: 25814582. This paper should be cited.

4. The authors have generated a great tool with the *T. cruzi* genome-wide library for YSD. However, it is very difficult to interpret the results from this screen, from the information given here and the very small sample size. No information is given on where the samples were collected, even if they are de-identified the region where they were collected should be known and is important. No ethics statement is included for the use of patient samples; it is unclear if the healthy controls are from endemic or non-endemic countries – ideally both should be included. There is no discussion on the alternative explanations for limited antibody recognition beyond strain diversification such as host immune response, PTMs, bias in the library...

Additionally, 5 infected samples and 2 control is a very limited sample size especially given the wide strain diversification in *T. cruzi*. A power calculation would ideally be included to determine whether the findings are statistically valid. The statements made regarding this experiment are too speculative and should be toned down throughout the paper.

Minor points:

- In general, we saw some discrepancies with the figure and supplementary material references in the text. For example, line 83, should this be Fig S3?
- Typo: 197: should say 'identity' not 'identify'.
- The paragraph from line 184-186 should be rephrased. It's challenging to follow and as it stands suggests antigenic variation in *T. brucei* is not stochastic. This is not correct.
- The percentage of infected cells for the first generation (G1) in Fig. 4E seems quite high. Looking at the literature for different strains, some with a higher virulence, the percentages of infection are more in tune with what is reported for G2 and G3. Could the authors explain this.
- In Figure 4B, it is not possible to understand the 1700-fold change in TS expression. Is this correct? Should it be 4C and could you include a description of the key in the figure legend – is this fold change?

(Remarks on code availability)

My expertise do not included being able to review this code.

Version 1:

Reviewer comments:

Reviewer #1

(Remarks to the Author)

The authors have undertaken a thorough revision of the manuscript, which, in my view, has substantially improved the work and demonstrates significant effort on their part. The revised manuscript has addressed some essential concerns raised during the first round of review. The authors improved the description of the central objective, moderated prior overstatements (such as the use of the term "stochastic"), and incorporated additional references to previous genomic studies. These are valuable improvements.

However, several significant issues remain unresolved, particularly in the interpretation of the genomic findings, the treatment of multigene families as a uniform functional group, and the incomplete adoption of established frameworks for *T. cruzi* genome organization.

Major Comments

1. Overstatement of novelty in genome organization and MGF distribution.

Despite the inclusion of some additional references and improved contextualization, the manuscript still presents several aspects of the genome analysis—such as the chromosomal distribution of multigene families (MGFs)—as novel findings. However, high-resolution assemblies and genome-wide analyses of *T. cruzi* structure, including telomere-to-telomere chromosomal resolution, compartmentalization (core, disruptive, and subtelomeric), and MGF distribution, have already been reported in recent literature (see below).

Rather than presenting these features as novel, the authors should reframe their findings as strain-specific confirmations or extensions of previously reported observations. The manuscript would benefit from emphasizing what is truly original in this study—such as the application of proteomic and immunological approaches in the Sylvio X10 background—rather than

reiterating, or presenting as new, features of *T. cruzi* genome architecture that are already well established.

Examples:

- Lines 14-16: “The genome reveals accurate organization of multigene families of virulence genes, either in expanded clusters or scattered throughout the chromosomes.”

- Lines 74-75: “The assembled genome defined the extent and locations of MGF genes, typically organized as gene clusters or spread throughout the chromosomes”

- Line 87 (“High-resolution genome assembly reveals chromosome-level multigene family organization”):

The use of 'reveals' implies novelty, which is not the case. The organization of these multigene families—both in expanded clusters and dispersed across chromosomes—has been extensively described in previous *T. cruzi* genome assemblies. This kind of overstatement diminishes the scientific accuracy of the manuscript.

Line 249: “Previous genome assembly of this and other strains contained hundreds of scaffolds, limiting the ability to resolve the MGF genome organization”.

This is not correct. The genomes of the strains Tulahuen (doi: 10.1093/g3journal/jkae076), Dm25 (doi: 10.1038/s41598-024-52449-x), and Dm28c (doi: 10.1101/2025.03.27.645724) strains were not fragmented into hundreds of scaffolds. Additionally, in all three assemblies, the genome organization of the MGFs was successfully clarified.

2. Avoidance of established genome compartment terminology despite extensive precedents.

This remains a significant concern: the authors continue to overlook the well-established conceptual framework that describes the *T. cruzi* genome as organized into core and disruptive compartments, despite citing studies that define these terms. This avoidance undermines the analysis of genome structure and contributes to the functional conflation of gene families. The core/disruptive framework is now standard in *T. cruzi* genomics, offering a powerful model that explains differences in gene density, GC content, length of directional gene clusters, and chromatin organization. First defined in 2018 (doi: 10.1099/mgen.0.000177), multiple research groups have widely confirmed this compartmentalization, and this terminology is now standard in the field. It allows for precise interpretation of chromosome structure, multigene family distribution, chromatin regulation, and evolutionary dynamics. However, the manuscript avoids using this vocabulary, often substituting vague phrases like “housekeeping gene clusters” (see below).

This omission is substantial, as it leads to conceptual confusion, which in turn weakens genomic interpretations.

Some references confirming structural and/or functional compartmentalization into core and disruptive:

- doi: 10.1099/mgen.0.000177: Original definition of core and disruptive genomic compartments [PMID: 29708484].

- doi: 10.1186/s13072-022-00450-x: Demonstrates chromatin openness differences between compartments [PMID: 35650626].

- doi: 10.1128/mbio.00319-24: Correlates replication origin mapping with compartment locations [PMID: 38441981].

doi: 10.1186/s12864-025-11384-5: MASP gene classification study distinguishing core vs. disruptive localization.

doi: 10.1101/2025.03.27.645724: Complete molecular karyotype of *T. cruzi* (PREPRINT).

doi: 10.1038/s41564-023-01483-y: Uses 3D genome interaction mapping to delineate core and disruptive nuclear domains.

- doi: 10.7554/eLife.105822.1: *T. cruzi* single-cell analysis and correlation with compartments (e.g., Figure 3; PREPRINT)

doi: 10.1016/j.pt.2025.04.008: Recent review summarizing recent *T. cruzi* genome organization.

This body of literature demonstrates that the field has converged on a shared vocabulary and conceptual framework.

However, the authors did not revise or reframe these conceptual aspects as suggested during the review process.

3. Oversimplification of distinct multigene families.

This is one negative consequence of the issue discussed above. The manuscript repeatedly groups all MGFs—TS, mucins, MASPs, gp63, DGF-1, and RHS—as though they share common roles, such as immune evasion (example below). This represents a serious conceptual error. While TS, mucins, and MASPs are well-characterized GPI-anchored surface proteins and known antigens, DGF-1 and RHS are neither surface-expressed nor relevant antigens. They are not even described as virulence factors. These gene families likely have unrelated roles within the cell.

The text must differentiate these functionally distinct groups throughout the manuscript, both when interpreting expression data and when drawing conclusions about host–parasite interactions and antigenic variation. In fact, DGF-1 and RHS do not belong to the disruptive compartment, as established in prior studies (2018, doi:10.1099/mgen.0.000177; 2025, doi:10.1101/2025.03.27.645724).

An illustrative example appears in line 81:

“Their antibody-recognition sites exhibited limited conservation, suggesting that sequence diversity and expression variation might contribute to antibody evasion. The data reveal MGFs' genomic organization, stage-specific expression, and variation in trypomastigotes, indicating that their sequence diversity and variation might contribute to heterogeneous parasite populations and a potential role in immune evasion.”

As mentioned, by grouping all MGFs together, the authors conflate functionally distinct gene families, leading to misleading and inaccurate interpretations. For example, the phrase 'potential role in immune evasion' is appropriate for TS and mucins (and is in fact well established), but not for DGF-1 and RHS, which are primarily expressed in epimastigotes and amastigotes—not trypomastigotes—and have no demonstrated role in immune modulation.

4. Misuse of the term “housekeeping genes” to describe the core compartment

Regarding the core compartment—characterized by its high synteny with other trypanosomatids and thus of clear functional importance—the authors eschew this definition and instead refer to it as a region rich in “housekeeping genes,” which is a conceptual mistake. Most core genes in *T. cruzi* encode hypothetical proteins and cannot be classified as housekeeping genes. Additionally, this term (core compartment) originates from a broader and well-established concept in comparative genomics: the “core genome” refers to the conserved, syntenic part of the genome shared among related organisms. It is not a term unique to trypanosomes but one that applies across the entire tree of life, from bacteria to higher eukaryotes. While housekeeping genes are certainly enriched in core regions, the term “core” encompasses broader features of genome architecture, including sequence conservation, chromatin state, and structural stability.

5. Chromosomes and Haplotypes

- PacBio HiFi sequencing currently offers the highest resolution available for genome assembly. In *T. cruzi*, it has previously been applied to two strains, yielding near-complete (doi: 10.1038/s41598-024-52449-x) or complete (doi: 10.1101/2025.03.27.645724) chromosome-level assemblies, which enabled the resolution of the full karyotype. One important feature of this sequencing technology is its ability to separate haplotypes, which is highly valuable in multiple contexts. Once the authors corrected the misassembly involving chromosome 22, it is surprising that they report haplotype separation only for this chromosome. In fact, with a proper analysis of the assembly, it should be possible to resolve haplotypes for all chromosomes. This raises the possibility of a technical or analytical issue that warrants further examination. One potential consequence of this misanalysis is that the reported copy numbers of multigene families may refer to the combined diploid genome, rather than per haploid genome, which should be clarified.

- On what basis are the chromosomes numbered? This should be reconsidered. The 2025 preprint uses chromosome length as the criterion, and we suggest that the same approach be adopted here for consistency.

- The authors should provide a rationale for why the remaining chromosomes could not be fully resolved, especially considering the high sequencing depth available.

6. Discussion Lacks Depth

Although the authors have reformulated the manuscript in response to reviewer suggestions, the Discussion section still lacks depth and includes several statements that are either self-evident or overly generic. For example:

- “Moreover, MGFs were expressed from various loci across all chromosomes, as detected by RNA-seq and proteomics, indicating no specific genomic site for their expression.”
If MGFs are distributed across multiple chromosomes, this observation is expected and does not constitute a novel finding.

- “The MGF genomic organization may facilitate transcription by RNA polymerase II, resulting in constitutive polycistronic transcription and the production of numerous transcripts within a cell.”
This reflects a general feature of genome organization in trypanosomatids and is not specific to MGFs. As such, it adds little to the interpretation of the results.

- “However, post-transcriptional levels of expression control may determine which transcripts are translated. This is consistent with the 3'-UTRs of mucins and TSs regulating their expression.”
This is a well-established and widely accepted concept in *T. cruzi* gene regulation, and its inclusion here without further analysis or specificity renders it superficial.

The discussion would benefit from deeper analysis, a more straightforward integration of the proteomic data with the genomic context, and a more focused examination of what is truly novel or unexpected in the findings.

Reframing the focus of the manuscript around proteomic and immunological findings

Given the growing number of high-resolution *T. cruzi* genome assemblies—including recent telomere-to-telomere studies—the primary strengths and novelties of this manuscript likely lie in its quantitative proteomics and antigen discovery components. These are valuable datasets that complement the genomic context, providing new insights into stage-specific protein expression and immune recognition. The authors should consider re-centering the manuscript around these contributions, clearly distinguishing between genomic features that are confirmed versus those that are newly discovered, and emphasizing the integrative nature of their approach to parasite biology. In particular, a comparative analysis of the antigenic properties of different MGF members would be a valuable addition, as would a detailed examination of differential expression patterns within each gene family.

Recommendation

The manuscript has improved considerably and includes valuable data. However, I recommend minor revision to address the conceptual conflation of MGFs, adopt standard genome terminology, properly contextualize the genomic findings in light of recent literature, and adjust the manuscript's focus to highlight its most substantial contributions. These changes will ensure the paper is accurate, conceptually rigorous, and impactful within the *T. cruzi* field.

(Remarks on code availability)

It is right.

Reviewer #2

(Remarks to the Author)

The authors have made substantial changes to the manuscript, addressing all the concerns that I raised. It reports new and interesting findings, providing an important advance in understanding of genome organisation and gene expression in *Trypanosoma cruzi*.

(Remarks on code availability)

n/a

Reviewer #3

(Remarks to the Author)

All comments have been dealt with. I have no further suggestions.

(Remarks on code availability)

Version 2:

Reviewer comments:

Reviewer #1

(Remarks to the Author)

I acknowledge the authors' efforts in addressing the suggestions, which should contribute to clarifying the work and highlighting the value of the results obtained. In my opinion, this manuscript is now close to being acceptable for publication in Nature Communications.

My only concern, which I consider essential to revise, relates to the transcription of disruptive genes, particularly in the context of the following statements:

Line 282 (added in the last revision):

"The MGF genes are likely transcribed constitutively, a hallmark of trypanosomatids, resulting in numerous transcripts within a cell, but controlled post-transcriptionally, given that single-cell analyses suggest only a few MGF genes are expressed per cell."

Line 302:

"This aligns with data showing that most, if not all, TSs are transcribed; (...)"

In reference 52 (Díaz-Viraqué et al.), a deep transcriptome analysis was performed, and the authors reported low or undetectable levels of expression for disruptive genes (TS, MASP, mucins):

"By contrast, the disruptive compartment, which in some cases comprises almost entire chromosomes, is enriched in low-expressed genes or genes with undetectable RNA levels (Fig. 2a)."

Overall, transcription of disruptive genes is low to very low. Reference 52 further shows that only specific disruptive genes are upregulated in trypomastigotes, and it would be valuable to assess whether this is also reflected at the proteomic level using the data generated in this work. This last point is a suggestion rather than a requirement.

What is essential, however, is to avoid referring to the transcription of disruptive genes as constitutive in general terms, since the evidence indicates that their expression is low or undetectable.

(Remarks on code availability)

The reviewer's comments are in black, and the author's answers are in blue.

Reviewer #1 (Remarks to the Author):

This manuscript presents a chromosome-level assembly of the *T. cruzi* Sylvio X10 strain (TcI) genome, utilizing PacBio HiFi long reads integrated with chromatin conformation capture via Nanopore-based Pore-C. This approach results in a 38.1 Mb assembly with an N50 of 1.3 Mb and 31 assembled chromosomes (27 of which are complete end-to-end). Achieving telomere-to-telomere contiguity for most chromosomes represents a significant technical strength, given the challenges associated with assembling *T. cruzi* multigene families. The authors further validate the accuracy of the assembly through depth-based ploidy analysis, revealing that most chromosomes are diploid with a few aneuploidies (one haploid and one triploid chromosome). Overall, the genome assembly approach is rigorous and provides a solid foundation. Next, the authors combine TMT-based quantitative proteomics with RNA-seq to assess gene expression across the life stages, identifying hundreds of differentially expressed proteins. As expected, trans-sialidases, mucins, and MASP are upregulated in cell-derived trypomastigotes. In contrast, the DGF-1 and RHS families are enriched in epimastigotes or amastigotes. While the proteomic resolution for highly similar gene family members is limited, integration with RNA-seq and a targeted nanopore-based “MGF-seq” confirmed the broad expression of MGF genes. A bimodal distribution of TS transcripts suggests post-transcriptional regulation, though detection limits in proteomics may still be a concern. Overall, the methods are robust and innovative, particularly in the temporal tracking of gene expression over serial infections. Finally, the Yeast Surface Display (YSD) Immunoscreen is a highlight of the strategy, showcasing creativity in addressing the immunological relevance of the parasite's gene diversity. This manuscript presents relevant and original findings by combining cutting-edge genomic, proteomic, and immunoscreen approaches. However, several aspects of the study need to be revised, as detailed below.

Authors: We appreciate the level of depth provided by the reviewer's comments. The comments have been valuable and constructive and helped reshape the manuscript into a better version. Answers to each point are provided below.

Major comments

1- Overall, the manuscript presents a substantial amount of data. While the authors provide numerous observations and analyses, the work would benefit significantly from more precisely articulating the overarching scientific question or hypothesis being addressed. Currently, the central aim of the study remains implicit. It is essential for the authors to clearly state the main research question early in the manuscript and ensure that each section is structured to contribute directly to answering that question. Furthermore, each section should be more tightly integrated into the narrative, with improved transitions and logical flow.

Authors: To clarify the study's overarching goal, we included in the Introduction, lines 69-74: “*In this work, we aimed to investigate the genomic organization and pattern of expression of MGF genes in T. cruzi, with a particular focus on whether temporal variation in gene expression could diversify the parasite population. Moreover, we investigate how their variation affects host cell infection and host antibody interactions. Understanding the expression of virulent genes in T. cruzi and their host interaction might provide insights into parasite infection and persistence.*”. We also edited the text to improve each Results session's transitions, i.e., the beginning and end of each session (see text highlighted in yellow in the manuscript).

2- A major concern is that, throughout the manuscript, the authors seem to follow a narrative based on the “pre-long-read sequencing/Hi-C era,” relying on concepts that date back to the publication of the first draft genome of *T. cruzi* in 2005. At that time, the genome's highly repetitive nature, combined with the limitations of Sanger sequencing, resulted in a highly fragmented assembly. This fragmentation led to the concept of “multigene family (MGF) clusters,” as it was not feasible to infer higher-order genome organization. However, since the first long-read-based *T. cruzi* genome assemblies, it has become increasingly clear that the genome is much more structured than previously thought. A conserved core region has been identified, showing synteny with *T. brucei* and *Leishmania*. In contrast, a disruptive region—characteristic of *T. cruzi*—is enriched in large arrays of mucins, MASPs, and trans-sialidases (TS). These disruptive regions are GC-rich and span extensive, continuous tracts of the chromosomes. In fact, entire scaffolds can now be classified as mainly core, mainly disruptive, or mixed, whereas in the current manuscript, the authors refer to the presence of MGF tracts as if this were a novel observation. This interpretation needs substantial revision, considering the existing body of work (see references at the end of this point). Additionally, these compartments vary in their nucleosomal organization and chromatin structure, contributing to their spatial segregation into distinct nuclear territories—specifically, 3D compartments C and D. Moreover, the DGF-1 and RHS families have consistently been associated with subtelomeric regions (the first publications are at least from 2005), suggesting they may play unique functional roles. These gene families should be discussed separately rather than treated as part of a general multigene family group. Finally, the trans-sialidase gene family comprises at least eight groups of TS genes that have been described. One of these groups encodes catalytically active enzymes with bona fide TS activity, while the remaining groups are functionally diverse. This functional and structural classification must be clearly considered when discussing the TS family and their patterns of expression. As just one example, in the abstract, the sentence “The genome revealed multigene families of virulence genes accounting for ~70% of some chromosomes, organized in clusters or scattered through housekeeping genes” seems to present previously established findings as if they were novel. Several other statements throughout the manuscript describe well-known observations without citing prior literature, which may inadvertently convey a sense of novelty for findings that have already been reported. For instance: Line 98 – “In contrast, DGF-1 and RHS were enriched in subtelomeric

regions.” → This has been previously established. Lines 99–100 – “MGF sequences correlated with a high GC content and were rich in LINE/I-L1Tc retrotransposons and RHS.” → Not all MGFs are GC-rich, and RHS genes, being mainly subtelomeric, should not be grouped indiscriminately with the rest of the MGFs. g) Line 122 – “We identified a stage-specific pattern of MGF protein expression, characterized by variations in protein abundance across each stage. TSs, mucins, and MASPs were significantly upregulated in CTs.” Stage-specific expression of these multigene families has already been reported.. Line 125 – “whereas GP63 was up-regulated in AMs and CTs.” → Also previously described. A particularly important example appears in the discussion (lines 210–215): “*T. cruzi* MGF genes are typically dispersed throughout the genome in clusters that alternate with housekeeping genes. This is evident for TSs, mucins, MASPs, and GP63, and it may facilitate their expression, as this chromosome organization may position them near other transcribed sequences. Exceptions include the DGF-1 and RHS families, which are enriched in subtelomeric regions. Notably, ~70% of some chromosomes consisted of repeat sequences containing MGF genes, suggesting their expansion in these chromosomes.” This entire passage recapitulates well-established observations without citing any previous work. As noted above, these features have been known for some time, and the lack of attribution is a serious oversight that should be addressed.

Authors: We appreciate the reviewer's thorough evaluation of the work and valuable comments. The key points are addressed below:

a) Referencing previous work: We edited the text to acknowledge other studies, emphasizing past observations throughout the text. Note references 6-8, 38-40, 51. We agree with the reviewer that previous work indicated scaffolds with clusters of housekeeping genes (core), clusters of multigene families (disruptive), and a mix of both. However, our genome assembly resolved many ambiguities related to MGF gene organization as we assembled complete chromosomes. Some of the previous studies had gaps and hundreds of unresolved scaffolds (particularly for the strain we report), which limited, in some cases, the confidence in the observation. In the Discussion, lines 249-253 were included: “*Previous genome assembly of this and other strains contained hundreds of scaffolds^{6, 7, 8, 39, 40}, limiting the ability to resolve the MGF genome organization. The assembly of most chromosomes from telomere to telomere confirmed previous observations of MGF distribution and expanded the knowledge of their genome organization. The assembly represents an advance over previous versions of the Sylvio XI0/1 strain^{38, 39}, resolving the genome into 31 chromosomes...*”. Also, in lines 264-270: “*Exceptions include the DGF-1 and RHS families, with several genes enriched in subtelomeric regions and in agreement with previous observations^{6, 7, 8, 38, 39, 40}, and their genomic locations might result in differences in gene expression regulation. The size and distribution of MGFs also varied significantly across strains (Supplementary Fig. 6), also noted by others⁵¹ ...The MGF gene organization differs from the earlier reports of discrete MGF clusters in scaffolds^{38, 39, 40} and is well aligned with recent observations from high-resolution *T. cruzi* genome assemblies^{6, 8}.*”

b) Distribution of DFG-1 and RHS and GC content: There are 221 DGF-1 and 521 RHS genes in our assembled genome (See Table 1 in the manuscript). We found that some DFG-1 and RHS are subtelomeric, whereas others are non-subtelomeric. We considered the reviewer's comments and included in the Results, lines 120-123: “*In contrast, 177 DGF-1 and 378 RHS genes were enriched in subtelomeric regions, whereas 44 DGF-1 and 129 RHS genes were spread throughout MGF clusters in the core chromosome (Fig. 1A, D-E, Supplementary Fig. 7). This is consistent with previous observations of their enrichment in subtelomeres^{7, 8, 39, 40}, but highlights that they are not exclusive to subtelomeric regions.*”. We also noted in the Discussion, lines 264-265: “*Exceptions include the DGF-1 and RHS families, with several genes enriched in subtelomeric regions and in agreement with previous observations^{6, 7, 8, 38, 39, 40}, ...*”. Regarding GC content, we performed additional analysis and show that RHS GC content is ~50%, and DGF-1 is ~65%, regardless of their genomic location. We included this analysis as Supplementary Fig. 7, and noted in the Results, lines 125-127: “*MGF sequences typically correlated with a high GC content (~ 65%) and were rich in LINE/I-L1Tc retrotransposons (Fig. 1A, D-E), except for RHS sequences, which were not GC-rich (~50%) regardless of core or subtelomeric locations (Supplementary Fig. 7).*”

c) TS analysis: We considered the TS groups in the genome and gene expression analysis. With respect to the genome, Table 1 shows the number of TSs in each group, and the new circos plot (Fig. 1A) highlights the distribution of TS groups across chromosomes. Figs 3B and C show TS group expression analysis by proteomics, with Fig 3C showing the comparative expression of each group among stages. This analysis was expanded in Supplementary Fig. 9. This has also been extensively discussed in the text. We also included how antibodies react to various TS groups, Results, lines 230-232: “*TSs from all eight groups reacted with human antibodies, with 37.9% of TSs being group V, 16% group II, 8% group VI, 4.6% group VIII, ~2% groups I, III, or IV, and 27.5% being unclassified TSs.*”. We also included a discussion highlighting the significance of group analysis in the Discussion, lines 288-290: “*Based on amino acid sequence characteristics, TSs have been categorized into 8 distinct groups⁵⁰. TS groups II, V, and VIII are the most abundant in the Sylvio XI0 genome (Table 1), and all TS groups were upregulated in CTs compared to other stages. ...*”.

d) Protein expression analysis: We included references for well-known protein expression, e.g., references 24, 25, 28, 33, 41 and 43, some of which validated our dataset (e.g., GP82 in MTs, Amastin in AMs, TSs in CTs). See Results, line 152-154: “*In contrast, DGF-1 and RHS were upregulated in AMs, MTs, and EPs, whereas GP63 were up-regulated in AMs and CTs (Supplementary Fig. 9), in agreement with other observations^{24, 25, 28, 33, 43}.*”. Note that TMT-labelling is an innovative technology that allowed us to accurately quantify the expression of a large number of proteins and compare their abundances among stages, which was not done previously in other proteome studies. The quantitative approach helped us to capture variations in protein expression in trypanomastigotes after host cell infection, which was also not reported before.

3. Misuse of the Term “Stochastic” and Need for Reframing the Interpretation.

The use of the term stochastic in the manuscript is potentially misleading. The observed variability in gene expression reflects an underlying heterogeneous parasite population, as demonstrated by single-cell RNA-seq studies. These studies (<https://doi.org/10.1101/2024.10.01.616042>; doi: <https://doi.org/10.1101/2025.01.14.633000>) have shown that multigene family expression varies significantly from one parasite to another. This has important implications, requiring reconsideration of both the title and the discussion. Rather than framing the observed variability as stochastic, the authors should emphasize that expression heterogeneity in *T. cruzi* is a well-established phenomenon. What should be highlighted as a novel contribution of the present study is the observation that passage through cultured host cells already imposes selective pressure, leading to shifts in subpopulation composition. This suggests that multigene family expression can be modulated by intrinsic factors or environmental cues, even in the absence of immune selection. At that point in the discussion, the authors may suggest a stochastic element as a contributing factor, but it should not be presented as the central mechanism. For example, if this process were similar to VSG switching in *T. brucei*, where clonal variation is followed by immune-driven selection, one would not expect passage through cultured host cells to be sufficient to shift expression patterns. The fact that it does suggests a high capacity for intrinsic plasticity in *T. cruzi*. Importantly, dormancy could be the underlying cause of the observed shifts. A subpopulation may remain in a transcriptionally silent (or low-activity) state in a stochastic manner, and upon successive passages, different subpopulations may become dominant. This possibility provides a plausible mechanistic explanation and should be discussed more explicitly in the manuscript.

Authors: The term stochastic in the presented context refers to the gene selected to be expressed. For example, out of 1,000 TSs, the gene expressed by each cell in a population appears stochastic rather than programmed, resulting in a heterogeneous cell population. To avoid confusion, we have clarified this in the discussion, de-emphasized the term stochastic throughout the manuscript, and removed it from the title. We added to the Discussion, lines 298-299: “*In this context, the choice of the TS gene to be expressed is likely stochastic, resulting in a heterogeneous parasite population*”. We also included a text in the Discussion, lines 307-309, regarding additional interpretation for the potential cause of variation: “*Although the change in MGF expression seems linked to parasite development, other intrinsic or extrinsic factors could contribute to the observed gene expression changes. Given that only 0.2-7% of parasites are dormant during infection⁵⁶, dormancy is unlikely to be the underlying cause of variation, but could contribute to it.*”.

3- Concerning ploidy: I strongly suggest mapping a deep set of Illumina reads to increase the sensitivity of the analysis. In addition, the distinction between chromosomes 22 and 22b remains unclear. Chromosome 22 is described as monosomic, yet the figure also shows a chromosome labeled 22b, which appears to display a monosomic pattern as well. Clarification of the relationship between these two chromosomes is needed. Lines 83-84: “Sequence depth confirmed chromosome copy numbers and indicated segmental aneuploidy in chromosome 1.” It is unclear what the authors mean by “segmental aneuploidy.” Please revise the figures.

Authors: We obtained an average sequencing depth of 397× with PacBio HiFi data, which allowed a reliable ploidy estimation. Nevertheless, we included additional data for ploidy analysis using nanopore sequencing (depth of 70x), which corroborated the results well. The results were incorporated into Supplementary Fig. 2, and in Results, lines 96 and 101: “Ploidy analysis by depth (average 397x) revealed 29 diploid chromosomes... Sequence depth using nanopore sequencing (70x) corroborated chromosome copy numbers obtained by PacBio depth analysis (Supplementary Fig. 2)”. As for chromosome 22, it is diploid, but there is an insertion in one of the haplotypes. We corrected Fig. 1B to display the diploid chromosome and indicated haplotype differences (diagramed in Fig 1A). We noted in the Results, lines 94-96: “...we resolved the two haplotypes for chromosome 22, namely 22a and 22b. Haplotype 22a contains a 126,891 bp insertion, which includes *LITc* elements followed by *trans-sialidase* (*TS*) genes (Fig. 1A).” The variable depth for part of chromosome 1 suggests potential extrachromosomal elements.”.

4. The authors report transcriptomic differences across *T. cruzi* life cycle stages based on an MGF-enriched transcriptome. However, this approach does not eliminate the possibility of technical bias introduced by the enrichment process. It is unclear why a standard bulk RNA-seq was not performed in parallel to provide an unbiased, genome-wide expression profile. We strongly recommend that the authors compare the differential expression patterns they report at least with those obtained from other publicly available RNA-seq datasets, such as the bulk RNA-seq used to validate the single-cell RNA-seq atlas [doi: <https://doi.org/10.1101/2024.10.01.616042>]. This comparison would help validate the observed expression trends and determine whether they are consistent with prior observations or possibly skewed by the targeted approach. Clarifying this point is crucial, especially when interpreting gene family expression dynamics that are known to be highly variable and sensitive to both biological and technical factors.

Authors: The suggested data refers to a manuscript deposited in bioRxiv, but the data is not publicly available yet, and the manuscript has not been peer-reviewed. Hence, we obtained RNA-seq data from SRR2177699 and SRR2177698 (HiSeq Illumina) and SRR9202394 (Miseq Illumina) for the Sylvio strain cell culture-derived trypomastigotes. We found that both RNA-seq data sets showed multiple MGFs transcribed, and variable levels of MGFs were detected among the datasets, analogous to our observations with MGF-seq. We included the analysis in the Supplementary Figure 13. We noted the correlation in Results, lines 184-186: “*Analysis of two distinct RNA-seq confirmed multiple MGF genes expressed at different levels in the CT population (Supplementary Fig. 13), also noted by the proteomic analysis (Fig. 3).*”.

Minor comments.

1- While this deep sequencing approach is robust, it is necessary to consider whether highly repetitive gene family regions still pose assembly ambiguities. Indeed, the authors find 39 short unplaced scaffolds (6–68 kb) that likely represent segmental aneuploidy or divergent haplotype segments with multigene family duplications. These residual scaffolds suggest that some repeats or haplotype-specific sequences could not be perfectly resolved. I ask the authors to clarify if these unincorporated segments contain additional gene copies or allelic variants of virulence genes.

Authors: The scaffolds contain sequence variants, mostly hypothetical proteins and a few MGFs (6.9% of genes). We included Supplementary Table 1 showing the synteny relationship of scaffolds to chromosomes and sequence similarities (also noted in Supplementary Fig. 4). We clarified in the Results, lines 103-107: “*There were 39 short-length scaffolds ranging from 6 to 68 kb with a summed size of less than a tenth of the smallest chromosome. Synteny analysis revealed that scaffolds shared 70-98% similarity with chromosomal regions, indicating they represented haplotype differences encoding primarily hypothetical proteins and a few virulence-related MGF genes (~7% of genes) (Fig. 1C, Supplementary Fig. 4, Supplementary Table 1).*”

2- Regarding the above, it is essential to clarify whether multimapping reads still exist (as with short read sequencing) and how to handle them, or whether PaBioHiFi is sufficient to clearly distinguish highly homologous sequences coding for MGF.

Authors: We performed a de novo assembly and did not map the PacBio HiFi sequences to a previously assembled genome. The PacBio HiFi sequence N50 is 8,976, with a mean quality of 99.9% accuracy and a depth of 397x. Hence, the sequences can distinguish highly homologous sequences, including different MGF sequences. We included some of this information in the Methods, lines 361-362: “...resulting in 15,8 billion bases, 1,98 million reads, with an N50 of 8,976.”. Also in Results, lines 91-92: “*The assembled nuclear genome was 38.1 Mb, with an N50 of 1.30 Mb and the largest contig being 2.63 Mb.*”.

3- The phrase “...infection, suggesting epigenetic regulation...” is not supported by data and should be removed or rephrased to reflect its speculative nature. **Authors:** We removed the word epigenetic.

4-166: “The increased production of CTs after consecutive H9-C2 infections (Fig. 4D) may be due to increased AM proliferation fitness rather than selection of CTs for invasion.” This interpretation could be directly tested by quantifying intracellular amastigotes. The phrase “may be due to” should be avoided in the absence of supporting data.

Authors: We removed the sentence.

5- In the circos diagram, there is no clear rationale for grouping MASP, mucins, and GP63 together while treating TS separately. The grouping should be biologically justified or revised for consistency. **Authors:** The rationale was to separate the TS into its 8 classified groups, which would not be visible otherwise. We replaced the circos plot (Fig. 1A) to improve visualization, highlighting TS groups in different colours.

6- Several figures present schematic representations rather than actual results. For instance, the pipeline diagram (Fig. 1A) and the protein labeling schematic (Fig. 2B) do not convey original data and could be moved to supplementary material or replaced with concise textual descriptions. **Authors:** We believe the diagram helps less experienced readers to understand the experiment's design, some of which are complex, such as in Figs. 4 and 5.

7. These figures should be reformulated to reflect the distinction between core and disruptive compartments, which is central to current genome organization models in *T. cruzi*. **Authors:** The choice of descriptive terms is to improve clarity for a general reader audience, avoiding jargon. Core and disruptive are often associated with the 3D genome compartments and are still not fully adopted to describe assembled genomes. Nevertheless, we appreciate the terminology and acknowledge and clarify the terms throughout the text. For example, in Discussion, lines 261-262: “*T. cruzi* MGF genes are typically dispersed throughout the genome in clusters that alternate with housekeeping genes, also referred to as disruptive and core regions^{6,52}, respectively.”.

Reviewer #1 (Remarks on code availability):

The code availability allows to download and analyze all of the data.

Reviewer #2 (Remarks to the Author):

In this manuscript, Saavedra and colleagues seek to deepen understanding of the functions provided by the very extensive multigene families (MGFs) in the genome of *Trypanosoma cruzi*. The findings of the manuscript can be summarised as follows: a new genome assembly of *T. cruzi* Sylvio X10 strain, clarifying the number of chromosomes and level of aneuploidy; demonstration of life cycle changes in the expression of several MGFs at both the protein and RNA level; and demonstration of immune recognition of several MGF proteins in Chagas patients. These are valuable insights, where the improved genome assembly has allowed much greater clarity of MGF expression dynamics across the *T. cruzi* life cycle, and has allowed defined understanding of immune recognition of the

MGFs. If communicated carefully, the manuscript will be of value, but I'm afraid the title, abstract and results are worded in ways that exaggerate the findings, and should be changed, as explained below:

Authors: We appreciate the reviewer's evaluation of the work. We have changed the title, abstract, and manuscript text to avoid misinterpreting the data. We believe the changes, as indicated below, will address the raised concerns.

There are two problems with the presentation of the work:

1. The proteomics and RNAseq experiments rely on population-levels analysis, which mainly shows that different MGFs show maximal expression in different life cycle forms, and expression of these MGFs appears to be just a portion of the overall repertoire. Thus, it is inaccurate to say that there is 'stochastic variation in surface protein expression' (eg title), since the available data can be interpreted as programmed changes in MGF expression; have the authors asked if the limited expression of some MGFs (eg TSs in Fig.4) is due to expression of functional genes and non-expression of pseudogenes? Can they show changes in patterns of MGF parts of the families that might provide evidence of stochastic or programmed expression change? It is also not at all reasonable in the results (eg lines 130-131 and 153-156) to invoke comparisons with single-cell RNAseq data; this study cannot, in any way, determine if there is cell-to-cell MGF expression heterogeneity. (Please also note that the Inchausti et al paper (ref 33) was published in 2025, not 2005.). **Authors:** We appreciate the reviewer's point, and we agree that we have presented a population analysis, and at this point, the MGF expression data could be interpreted as programmed. We re-analyzed the MGF expression across multiple rounds of infection, but we did not detect a specific pattern pointing to programmed expression. Also, we detected the expression of functional genes rather than pseudogenes. Nevertheless, to prevent misinterpretation of our data, we have removed the word stochastic from the title and de-emphasized the term stochastic throughout the manuscript. We also included in the Discussion, lines 302-304: *"The analysis of MGF expression after consecutive rounds of parasite infection did not reveal a specific pattern supporting programmed MGF gene expression, although we cannot rule out this possibility."* As for the single-cell data, we edited the text for clarity and corrected the reference date, lines 159-161: *"Since single-cell data show transcription of different TSs in different cells⁴⁴, it is possible that the diversity of MGF proteins detected reflects a heterogeneous parasite population."*

2. The authors try to suggest that their data demonstrates MGF expression variation allows 'the invasion to multiple tissues'(abstract), and 'help[s] parasites avert antibody recognition' (abstract). Both conclusions significantly overstate the findings: the authors merely measure infection levels in two cultured host cell types, and demonstrate antibody responses to MGF proteins (but did not correlate changes in MGF expression with timing of antibody responses). These are reasonable speculations if limited to the discussion, but require significantly greater evidence to be presented as main conclusions, as they currently are (eg 'diversifies T. cruzi infection'; title). **Authors:** We have changed the abstract to avoid any suggestion of tissue infection and clarified in the text that we used three different cell lines. Moreover, we clarified that the antibody analysis might reflect antibodies against a heterogeneous parasite population and emphasized that a temporal analysis was not performed. We included in the Discussion, lines 328-330: *"The combination of TS sequence diversity and variation in their expression could help the parasite to escape host antibody recognition. Nevertheless, a temporal analysis of MGF gene expression variation and antibody production during infection would be necessary to establish a direct correlation."*

More specific comments:

3. Line 45. Provide references for parasite evasion of immunity in certain tissues.

Authors: Included in Introduction, lines 52-53: *"Persistence has been associated with the failure of CD8⁺-T cells to clear tissue infection^{14, 16, 17} and the expression of parasite molecules that counteract the tissue immune response^{18, 19, 20, 21.}"*

4. Introduction. For a non-specialist reader, it would aid clarity to outline the T. cruzi life cycle forms and where they are found.

Authors: We included in the Introduction, lines 43-48: *"In the insect, the parasite replicates as epimastigotes, differentiating into non-replicative metacyclic trypomastigotes. Metacyclics are released by the insect during a blood meal and infect the host cells. Inside the host cells, they differentiate into replicative amastigote forms. After several rounds of cell division, amastigotes differentiate into non-dividing trypomastigotes, which can spread in the bloodstream and infect new tissues, where they differentiate into amastigotes and replicate. The insect can take trypomastigotes during a blood meal and re-initiate the cycle."*

5. The description of the new genome assembly is very abbreviated and warrants some expansion. What are the implications of now defining the genome as comprising 32 chromosomes: what was known before; is this likely to be specific to this strain; can they summarise what was learned by comparing this genome assembly with several other strains (Fig.S4)?

Authors: We included the paragraph in the Discussion, lines 249-256: *"Previous genome assembly of this and other strains contained hundreds of scaffolds^{6, 7, 8, 39, 40}, limiting the ability to resolve the MGF genome organization. The assembly of most chromosomes from telomere to telomere confirmed previous observations of MGF distribution and expanded the knowledge of their genome organization. The assembly represents an advance over previous versions of the Sylvio X10/I strain^{38, 39}, resolving the genome into 31 chromosomes and providing a chromosome-level karyotype for a TcI Sylvio strain. While chromosome numbers might vary across DTUs, the 31 assembled chromosomes may help the analysis of other TcI strains. This improvement allows for accurate chromosomal structure and MGF distribution compared to previous assemblies."* And lines 266-274: *"The size and distribution of MGFs also varied significantly*

across strains (Supplementary Fig. S6), also noted by others⁵¹. Notably, ~70% of some chromosomes – especially 29 and 31 – consisted of repeat sequences containing MGF genes, suggesting their expansion in these chromosomes. The MGF gene organization differs from the earlier reports of discrete MGF clusters in scaffolds^{38, 39, 40} and is well aligned with recent observations from high-resolution *T. cruzi* genome assemblies^{6, 8}. In contrast, most housekeeping genes were conserved and syntenic with other strains. The annotation also enabled subtype-level classification of TSs and mucins, which helped with transcriptomic, proteomic, and antibody screening studies. The assembly provides a foundation for understanding genome architecture, antigenic diversity, and host-parasite interactions.”.

6. Is the pore-C data shown anywhere, and how does this compare with previous Hi-C work from the Robello lab?

Authors: The Pore-C data was used in the pipeline to assemble the contigs into chromosomes, and we did not perform a detailed comparison with Robello’s lab Hi-C, as it was not the scope of the study. Pore-C data (nanopore sequencing) typically has lower resolution than Hi-C (Illumina sequencing) but benefits from the detection of multiway contacts that are useful in genome assembly and detection of specific genome features (Zhong et al. 2023, Nat Commun; Open2C et al. 2024, PLoS Comput. Biol.). As a reference, we included a heatmap of the Pore-C interaction matrix in Supplementary Fig. 1 (mapped to our genome) and provided some additional data description.

7. Line 76. In what way can Nanopore sequence 'validate' the PacBio assembly?

Authors: The nanopore genome sequencing data from *T. cruzi* Sylvio X10 strain were included in Supplementary Fig. 2 to confirm full genome coverage and support the chromosomal ploidy analysis. Additionally, we included Supplementary Fig. 5 showing the number of nanopore reads, PacBio reads, and Illumina reads mapped to our assembly and compared to the previous Sylvio X10 genome version available in 2018 (Talavera-Lopez C, et al. 2021), demonstrating an increase in the reads mapping to our assembled genome. We noted in the Results, lines 101-102: “Sequence depth using nanopore sequencing (70x) corroborated chromosome copy numbers obtained by PacBio depth analysis (Supplementary Fig. 2)”, and lines 107-109: “Nanopore and Illumina sequencing were used to validate the assembly, demonstrating increased data mapping and quality scores for our genome compared to the previous assembly, with most reads mapping to the assembled chromosomes³⁸ (Supplementary Fig. 5)”.

8. Line 80. Please clarify why chromosomes 22 and 22b are considered different chromosomes and not merely alleles of a diploid chromosome (is a large insertion enough to make them distinct chromosomes?).

Authors: Thanks. There was a mistake in the Figure. We corrected the figure to show only one chromosome 22. We resolved two haplotypes for chromosome 22. We also clarified in the Results, lines 94-96: “...we resolved two haplotypes for chromosome 22, namely 22a and 22b. Haplotype 22a contains a 126,891 bp insertion, which includes LITc elements followed by TS genes (Fig. 1A).”.

9. Line 84, Fig.1C. Can the authors clarify how these data reveal segmental aneuploidy; can these contigs not represent extrachromosomal elements? Please also discuss how the ploidy/aneuploidy detected here compares with previous literature.

Authors: We further analyzed these regions and found that most represent haplotypes' differences. We included Supplementary Table 1, which shows the synteny analysis of scaffolds and chromosomes and their similarities. The scaffold sequences are very similar (70-98% similarity) to the assembled chromosomes. Supplementary Fig. 4 also refers to the analyses. We have revised the Results, lines 104-107: “Synteny analysis revealed that scaffolds shared 70-98% similarity with chromosomal regions, indicating they represent haplotype differences encoding primarily hypothetical proteins and a few virulence-related MGF genes (~7% of genes)”. Potential extrachromosomal elements exist for Chromosome 1 and segmental aneuploidy for some chromosomes, such as chromosome 10, as noted in Results, lines 98-99: “The variable depth for part of chromosome 1 suggests potential extrachromosomal elements, whereas some chromosomes might have segmental aneuploidies, e.g. chromosome 10 (Supplementary Fig. 2)”. We also added to the Discussion, lines 256-259: “Notably, most chromosomes were diploid, except chromosomes 1 and 30, and segmental aneuploidy was found in some chromosomes, e.g., chromosome 10. Previous studies on *T. cruzi* also reported chromosome segmental aneuploidies⁸; however, low-resolution assemblies can be influenced by the resolution of repetitive regions, leading to read collapse and inaccuracies in ploidy estimations. Ploidy differences can also be due to differences among strains. ”

10. Line 130 is a good example of over-interpreting the data: 'RNA-seq and proteomic analysis confirmed the expression of large subsets of MGF genes from multiple chromosome loci (Fig. 3E, Fig. S8). This agrees with single-cell data suggesting transcription of TSs from multiple loci in different cells (33). The diversity of MGF proteins expressed likely reflects heterogeneity within the parasite population.' There is no comparison made between these data and scRNAseq, and all these data can be interpreted as programmed expression of a selection of MGFs across the cell cycle.

Authors: We corrected the statement; see the answer to Q1.

Reviewer #3 (Remarks to the Author):

Stochastic variation in surface protein expression diversifies *Trypanosoma cruzi* infection

In this study by Saavedra et al, the authors describe variation in multigene families in *T. cruzi* and how this may aid parasite persistence in the host. By integrating high-resolution genome assembly, proteomics, and genome-wide yeast surface display, they

argue that variation in the expression of multigene families facilitates host cell invasion and to evade hosts immune response and how stochastic variation in the expression of these MGF contributes to the diversification of parasite populations. Overall, this is a well-executed and insightful study that advances our understanding of how *T. cruzi* leverages gene family variability to sustain chronic infection. **Authors:** We appreciate the valuable and constructive comments from the reviewer, which helped improve the manuscript. Answers to each point are provided below.

Main points:

1. The inclusion of a short paragraph in the introduction on the *T. cruzi* life cycle and development /expression of MGF would benefit the none-specialist reader.

Authors: We included in the Introduction, lines 43-48: “*In the insect, the parasite replicates as epimastigotes, differentiating into non-replicative metacyclic trypomastigotes. Metacyclics are released by the insect during a blood meal and infect the host cells. Inside the host cells, they differentiate into replicative amastigote forms. After several rounds of cell division, amastigotes differentiate into non-dividing trypomastigotes, which can spread in the bloodstream and infect new tissues, where they differentiate into amastigotes and replicate. The insect can take trypomastigotes during a blood meal and re-initiate the cycle.*”. Paragraph 3 of the Introduction also explains the many surface proteins of *T. cruzi* and their function, and specifics for each stage are also in Results, lines 144-145: “*The data included known stage-specific markers, such as GP82 and GP90 in MTs²⁸, TSs in CTs^{29, 30}, amastin in AMs⁴¹.*”.

2. There are several studies that report high-resolution sequencing and genome assembly of *T. cruzi* strains, including Sylvio X10, yet not referenced. Included in the discussion there should be a comparison of the assembly done here with published studies. These include:

- Wang W, Peng D, Baptista RP, Li Y, Kissinger JC, Tarleton RL (2021) Strain-specific genome evolution in *Trypanosoma cruzi*, the agent of Chagas disease. *PLoS Pathog* 17(1): e1009254. <https://doi.org/10.1371/journal.ppat.1009254>
- Callejas-Hernández, F., Rastrojo, A., Poveda, C. et al. Genomic assemblies of newly sequenced *Trypanosoma cruzi* strains reveal new genomic expansion and greater complexity. *Sci Rep* 8, 14631 (2018). <https://doi.org/10.1038/s41598-018-32877-2>
- Franzén O, Ochaya S, Sherwood E, Lewis MD, Llewellyn MS, Miles MA, Andersson B. Shotgun sequencing analysis of *Trypanosoma cruzi* I Sylvio X10/1 and comparison with *T. cruzi* VI CL Brener. *PLoS Negl Trop Dis*. 2011 Mar 8;5(3):e984. doi: 10.1371/journal.pntd.0000984. PMID: 21408126; PMCID: PMC3050914
- Díaz-Viraqué F, Chiribao ML, Libisch MG, Robello C. Genome-wide chromatin interaction map for *Trypanosoma cruzi*. *Nat Microbiol*. 2023 Nov;8(11):2103-2114. doi: 10.1038/s41564-023-01483-y. Epub 2023 Oct 12. PMID: 37828247; PMCID: PMC10627812.
- Dean AAC, Berná L, Robello C, Buscaglia CA, Balouz V. An algorithm for annotation and classification of *T. cruzi* MASP sequences: towards a better understanding of the parasite genetic variability. *BMC Genomics*. 2025 Feb 24;26(1):194. doi: 10.1186/s12864-025-11384-5. PMID: 39994548; PMCID: PMC11852901.

Authors: We included the relevant references, note references 6-8, 38-40, and 51. We also included in the Discussion, lines 249-271: “*Previous genome assembly of this and other strains contained hundreds of scaffolds^{6, 7, 8, 39, 40}, limiting the ability to resolve the MGF genome organization. The assembly of most chromosomes from telomere to telomere confirmed previous observations of MGF distribution and expanded the knowledge of their genome organization. The assembly represents an advance over previous versions of the Sylvio X10/1 strain^{38, 39}, resolving the genome into 31 chromosomes ... This improvement allows for accurate chromosomal structure and MGF distribution compared to previous assemblies.*” ... “*The size and distribution of MGFs also varied significantly across strains (Supplementary Fig. 6), also noted by others⁵¹. Notably, ~70% of some chromosomes – especially 29 and 31 – consisted of repeat sequences containing MGF genes, suggesting their expansion in these chromosomes. The MGF gene organization differs from the earlier reports of discrete MGF clusters in scaffolds^{38, 39, 40} and is well aligned with recent observations from high-resolution *T. cruzi* genome assemblies^{6, 8}. In contrast, most housekeeping genes were conserved and syntenic with other strains. ...*”.

3. The MGF-seq is good idea, however it follows the same principle as VSG-seq from Mugnier MR, et al. *Science*. 2015 PMID: 25814582. This paper should be cited. **Authors:** Done.

4. The authors have generated a great tool with the *T. cruzi* genome-wide library for YSD. However, it is very difficult to interpret the results from this screen, from the information given here and the very small sample size. No information is given on where the samples were collected, even if they are de-identified the region where they were collected should be know and is important. No ethics statement is included for the use of patient samples; it is unclear if the healthy controls are from endemic or non-endemic countries – ideally both should be included. There is no discussion on the alternative explanations for limited antibody recognition beyond strain diversification such as host immune response, PTMs, bias in the library... Additionally, 5 infected samples and 2 control is a very limited sample size especially given the wide strain diversification in *T. cruzi*. A power calculation would ideally be included to determine whether the findings are statistically valid. The statements made regarding this experiment are too speculative and should be toned down throughout the paper.

Authors: The sera is a pool of naturally infected patients' migrants from Latin America (Bolivia). Note reference 73, Golizeh M, et al. 2022, Heliyon, has detailed information on the cohort. The information has been included in the manuscript, Methods, lines 526-530,

and the reference is included. “Sera was kindly provided by Dr. Momar Ndao (McGill University Health Centre) from patients with Chagas disease or healthy individuals, who were Latin American migrants from Bolivia, with an age range of 28-56 years⁷³. Written informed consent was obtained from all participants⁷³. Serological tests were used to confirm the status of patients and control sera⁷³. Since we are screening for antibodies from pooled sera and not comparing patients' sera, the power analysis indicated is not possible nor adequate. Statistical analysis of library enrichment (fold-change and *p*-values) with patients' sera was included in Figure 5C. We included a paragraph in the Discussion addressing other potential limitations as indicated by the reviewer. See lines 320-: “Alternatively, it could also reflect biases in the host immune response to some parasite proteins or differences between post-translational modifications from yeast and parasites, which could prevent antibody recognition of some yeast-expressed proteins. The limited detection of parasite MGFs is unlikely to be YSD library bias, as the library size is 30-fold of the parasite genome, with an average of 270 clones/gene, and predicted ~250,000 polypeptides⁴⁸”. We edited the Discussion to avoid misinterpretation of the data, e.g. lines 327-330: “The combination of TS sequence diversity and variation in their expression could help the parasite to escape host antibody recognition. Nevertheless, a temporal analysis of MGF gene expression variation and antibody production during infection would be necessary to establish a direct correlation.”

Minor points:

- In general, we saw some discrepancies with the figure and supplementary material references in the text. For example, line 83, should this be Fig S3?. **Authors:** Thanks, we corrected this.

- Typo: 197: should say ‘identity’ not ‘identify’. **Authors:** Corrected.

- The paragraph from line 184-186 should be rephrased. It’s challenging to follow and as it stands suggests antigenic variation in *T. brucei* is not stochastic. This is not correct. **Authors:** I believe the reviewer refers to the last paragraph of the Discussion, lines 336-337. We edited for clarity to avoid assumptions that *T. brucei* antigenic switch is not stochastic. “*In this scenario, the MGF variation in T. cruzi may not be distinct from that in T. brucei, which is highly immunogenic and switches to evade host antibody detection*⁵⁹.”

- The percentage of infected cells for the first generation (G1) in Fig. 4E seems quite high. Looking at the literature for different strains, some with a higher virulence, the percentages of infection are more in tune with what is reported for G2 and G3. Could the authors explain this. **Authors:** An explanation might be the more homogeneous population in Generation 1, as shown in Fig. 4B and Supplementary Fig. 14C, which could favour invasion of the cell lines via parasite surface protein and host surface receptor interaction. As the population become more heterogeneous, with respect to MGF expression, surface protein changes may affect interactions with host receptors, decreasing the rate of infection. We discussed this in Results, lines 201-207: “*There was a notably high rate of invasion with the first generation of CTs (G1) (Fig. 4E), which correlated with their lower diversity in MGFs expressed (Fig. 4B, Supplementary Fig. 14). This suggests a more homogeneous parasite population at G1, perhaps favouring surface prote in engagement with host cell receptors for invasion, whereas the increased MGF diversification in subsequent generations might have affected invasion efficiency to specific cell types.*”

- In Figure 4B, it is not possible to understand the 1700-fold change in TS expression. Is this correct? Should it be 4C and could you include a description of the key in the figure legend – is this fold change? **Authors:** We have corrected it to clarify it. We also added a description to the legend, i.e. log₂ scale of normalized read counts for Fig. 4B (line 873) and linear scale of protein relative abundance for Fig. 4C (line 875).

Reviewer #3 (Remarks on code availability):

My expertise do not included being able to review this code.

Author's response in blue, reviewers' comments in black.

REVIEWER COMMENTS

Reviewer #1 (Remarks to the Author): The authors have undertaken a thorough revision of the manuscript, which, in my view, has substantially improved the work and demonstrates significant effort on their part. The revised manuscript has addressed some essential concerns raised during the first round of review. The authors improved the description of the central objective, moderated prior overstatements (such as the use of the term "stochastic"), and incorporated additional references to previous genomic studies. These are valuable improvements. However, several significant issues remain unresolved, particularly in the interpretation of the genomic findings, the treatment of multigene families as a uniform functional group, and the incomplete adoption of established frameworks for *T. cruzi* genome organization.

Major Comments

Reviewer: 1. Overstatement of novelty in genome organization and MGF distribution. Despite the inclusion of some additional references and improved contextualization, the manuscript still presents several aspects of the genome analysis—such as the chromosomal distribution of multigene families (MGFs)—as novel findings. However, high-resolution assemblies and genome-wide analyses of *T. cruzi* structure, including telomere-to-telomere chromosomal resolution, compartmentalization (core, disruptive, and subtelomeric), and MGF distribution, have already been reported in recent literature (see below). Rather than presenting these features as novel, the authors should reframe their findings as strain-specific confirmations or extensions of previously reported observations. The manuscript would benefit from emphasizing what is truly original in this study—such as the application of proteomic and immunological approaches in the Sylvio X10 background—rather than reiterating, or presenting as new, features of *T. cruzi* genome architecture that are already well established. Examples:
- Lines 14-16: “The genome reveals accurate organization of multigene families of virulence genes, either in expanded clusters or scattered throughout the chromosomes.” **Authors:** The sentence in the Abstract, lines 15-16, was modified for: “*The genome provides accurate organization of multigene family genes, confirming their distribution in expanded clusters or scattered throughout the chromosomes.*”. We also indicated the results is strain specific throughout the document.

- Lines 74-75: “The assembled genome defined the extent and locations of MGF genes, typically organized as gene clusters or spread throughout the chromosomes” **Authors:** We rephrase as: “*The assembled genome highlights the extent and locations of MGF genes, typically organized as gene clusters, also called disruptive compartment, or spread throughout the chromosomes*”; see lines 75-76.

- Line 87 (“High-resolution genome assembly reveals chromosome-level multigene family organization”): **Authors:** The sentence was modified for: “*High-resolution genome assembly results in chromosome-level gene organization*”, see line 89.

The use of 'reveals' implies novelty, which is not the case. The organization of these multigene families—both in expanded clusters and dispersed across chromosomes—has been extensively described in previous *T. cruzi* genome assemblies. This kind of overstatement diminishes the scientific accuracy of the manuscript. **Authors:** The term “*reveals*” was revised as per reviewer’s suggestion.

Line 249: “Previous genome assembly of this and other strains contained hundreds of scaffolds, limiting the ability to resolve the MGF genome organization”. This is not correct. The genomes of the strains Tulahuén (doi: 10.1093/g3journal/jkae076), Dm25 (doi: 10.1038/s41598-024-52449-x), and Dm28c (doi: 10.1101/2025.03.27.645724) strains were not fragmented into hundreds of scaffolds. Additionally, in all

three assemblies, the genome organization of the MGFs was successfully clarified. **Authors:** The sentence was deleted.

Reviewer: 2. Avoidance of established genome compartment terminology despite extensive precedents. This remains a significant concern: the authors continue to overlook the well-established conceptual framework that describes the *T. cruzi* genome as organized into core and disruptive compartments, despite citing studies that define these terms. This avoidance undermines the analysis of genome structure and contributes to the functional conflation of gene families. The core/disruptive framework is now standard in *T. cruzi* genomics, offering a powerful model that explains differences in gene density, GC content, length of directional gene clusters, and chromatin organization. First defined in 2018 (doi: 10.1099/mgen.0.000177), multiple research groups have widely confirmed this compartmentalization, and this terminology is now standard in the field. It allows for precise interpretation of chromosome structure, multigene family distribution, chromatin regulation, and evolutionary dynamics. However, the manuscript avoids using this vocabulary, often substituting vague phrases like “housekeeping gene clusters” (see below). This omission is substantial, as it leads to conceptual confusion, which in turn weakens genomic interpretations. Some references confirming structural and/or functional compartmentalization into core and disruptive: - doi: 10.1099/mgen.0.000177: Original definition of core and disruptive genomic compartments [PMID: 29708484]. - doi: 10.1186/s13072-022-00450-x: Demonstrates chromatin openness differences between compartments [PMID: 35650626]. - doi: 10.1128/mbio.00319-24: Correlates replication origin mapping with compartment locations [PMID: 38441981]. doi: 10.1186/s12864-025-11384-5: MASP gene classification study distinguishing core vs. disruptive localization. doi: 10.1101/2025.03.27.645724: Complete molecular karyotype of *T. cruzi* (PREPRINT). doi: 10.1038/s41564-023-01483-y: Uses 3D genome interaction mapping to delineate core and disruptive nuclear domains. - doi: 10.7554/eLife.105822.1: *T. cruzi* single-cell analysis and correlation with compartments (e.g., Figure 3; PREPRINT) doi: 10.1016/j.pt.2025.04.008: Recent review summarizing recent *T. cruzi* genome organization. This body of literature demonstrates that the field has converged on a shared vocabulary and conceptual framework. However, the authors did not revise or reframe these conceptual aspects as suggested during the review process. **Authors:** The term “core” and “disruptive” regarding chromosome compartments has been incorporated into the manuscript. We included in the Fig. 1 D and E and figure legends, Table 1, and manuscript text. I counted the word “disruptive” 8 times and “core” 10 times in the manuscript, for example in lines 112-133.

Reviewer: 3. Oversimplification of distinct multigene families. This is one negative consequence of the issue discussed above. The manuscript repeatedly groups all MGFs—TS, mucins, MASPs, gp63, DGF-1, and RHS—as though they share common roles, such as immune evasion (example below). This represents a serious conceptual error. While TS, mucins, and MASPs are well-characterized GPI-anchored surface proteins and known antigens, DGF-1 and RHS are neither surface-expressed nor relevant antigens. They are not even described as virulence factors. These gene families likely have unrelated roles within the cell. The text must differentiate these functionally distinct groups throughout the manuscript, both when interpreting expression data and when drawing conclusions about host–parasite interactions and antigenic variation. In fact, DGF-1 and RHS do not belong to the disruptive compartment, as established in prior studies (2018, doi:10.1099/mgen.0.000177; 2025, doi:10.1101/2025.03.27.645724). **Authors:** There is a lack of functional studies on DGF-1 and RHS these proteins, thus we cannot rule out a potential virulence role. Our proteomic and RNA-seq analysis show that DGF-1 and RHS are expressed in infectious forms (Figs. 3 and 4), and their expression varies as TSs, MASPs, and Mucins, hence their description here. To clarify, we included in the Introduction, lines 37-39: “*MGFs such as TSs, mucins, MASPs are associated with T. cruzi pathogenesis*⁹ and thus are known virulence factors, whereas DGF-1 and RHS functions

remain unclear.” We also added (line 67) a sentence to indicate that virulence MGF genes refers to TSs, mucins, MASPs, and GP63.: “...*amino acid sequence diversity of virulence MGF genes (i.e., TSs, mucins, MASPs, GP63)...*”. Also in Discussion, lines 323-327: “*MGF diversity and variable expression may impact host immune recognition, specially for those MGF genes encoding virulence factors such as TSs, MASPs, mucins, and GP63. The functional role of RHS and DGF-1 remains unknown. Their gene expansion, diversity, and stage-specific expression suggest potential roles in parasite survival or differentiation, intracellular host interaction, parasite-insect vector interactions, or stage-specific regulatory processes that aids parasite survival and infection.*”. And lines 349-351: “*Although the functions of many MGF genes remain unknown, their diverse repertoire may have evolved, at least in part, to facilitate host cell invasion and evade immune responses, supporting tissue tropism and parasite persistence.*”

An illustrative example appears in line 81: “Their antibody-recognition sites exhibited limited conservation, suggesting that sequence diversity and expression variation might contribute to antibody evasion. The data reveal MGFs' genomic organization, stage-specific expression, and variation in trypomastigotes, indicating that their sequence diversity and variation might contribute to heterogeneous parasite populations and a potential role in immune evasion.” As mentioned, by grouping all MGFs together, the authors conflate functionally distinct gene families, leading to misleading and inaccurate interpretations. For example, the phrase 'potential role in immune evasion' is appropriate for TS and mucins (and is in fact well established), but not for DGF-1 and RHS, which are primarily expressed in epimastigotes and amastigotes—not trypomastigotes—and have no demonstrated role in immune modulation. **Authors:** The sentence (lines 81-86) has been revised to: “*Their antibody-recognition sites exhibited limited conservation, indicating that the sequence diversity and changes in expression of some MGF members (e.g., TSs, mucins, MASPs) may contribute to antibody evasion. The data confirm MGFs' genomic organization and reveal stage-specific expression and variation in trypomastigotes, indicating that sequence diversity and variation of some gene families might contribute to heterogeneous parasite populations and a potential role in immune evasion.*”

4. Misuse of the term “housekeeping genes” to describe the core compartment. Regarding the core compartment—characterized by its high synteny with other trypanosomatids and thus of clear functional importance—the authors eschew this definition and instead refer to it as a region rich in “housekeeping genes,” which is a conceptual mistake. Most core genes in *T. cruzi* encode hypothetical proteins and cannot be classified as housekeeping genes. Additionally, this term (core compartment) originates from a broader and well-established concept in comparative genomics: the “core genome” refers to the conserved, syntenic part of the genome shared among related organisms. It is not a term unique to trypanosomes but one that applies across the entire tree of life, from bacteria to higher eukaryotes. While housekeeping genes are certainly enriched in core regions, the term “core” encompasses broader features of genome architecture, including sequence conservation, chromatin state, and structural stability. **Authors:** The term "core" was included in the manuscript and HK was removed.

Reviewer: 5. Chromosomes and Haplotypes. - PacBio HiFi sequencing currently offers the highest resolution available for genome assembly. In *T. cruzi*, it has previously been applied to two strains, yielding near-complete (doi: 10.1038/s41598-024-52449-x) or complete (doi: 10.1101/2025.03.27.645724) chromosome-level assemblies, which enabled the resolution of the full karyotype. One important feature of this sequencing technology is its ability to separate haplotypes, which is highly valuable in multiple contexts. Once the authors corrected the misassembly involving chromosome 22, it is surprising that they report haplotype separation only for this chromosome. In fact, with a proper analysis of the assembly, it should be possible to resolve haplotypes for all chromosomes. This raises the possibility of a technical or

analytical issue that warrants further examination. One potential consequence of this misanalysis is that the reported copy numbers of multigene families may refer to the combined diploid genome, rather than per haploid genome, which should be clarified. - On what basis are the chromosomes numbered? This should be reconsidered. The 2025 preprint uses chromosome length as the criterion, and we suggest that the same approach be adopted here for consistency. - The authors should provide a rationale for why the remaining chromosomes could not be fully resolved, especially considering the high sequencing depth available. **Authors:** The primary objective of this study (noted in Introduction per request of this reviewer in the first round of revision – lines 70-74) was to investigate the dynamics of multigene families (MGFs) expression, rather than to produce a phased genome. The genome assembly used PacBio HIFI and Pore-C and generated at high resolution assembly to support comparative analyses of MGF gene content, structure, and expression across life stages, resulting in the reconstruction of complete chromosomes. Generating a phased genome is beyond the scope and the goal of this manuscript, and unnecessary for the analysis presented.

For chromosome 22, there was no miss-assembly. We just clarified its terminology in the Figure 1, instead of calling chromosome 22 and 22b, we called haplotype 22a and 22b. Our data is consistent with the two haplotypes for chromosome 22 as reported in the DM25 strain (DOI: 10.1038/s41598-024-52449-x), which shows high synteny with our assembled sequences. In the Results section, line 96, we state: “***Additionally, we resolved the previously described⁷ two haplotypes for chromosome 22, namely 22a and 22b***”. Depth analysis with Pacbio (397x) and nanopore (70x) sequencing further confirm the presence of both haplotypes. The remaining 39 scaffolds are very short (smaller than 60 kb). All assembled chromosomes show strong synteny with the DM25 reference chromosomes, supporting the completeness and structural accuracy of our assembly. Further resolving the phase of the genome is beyond the scope of this work, as the genome assembly (also stated above by this reviewer) is not the focus of this work.

Chromosome numbering was revised to follow the NGSEP (Next Generation Sequencing Experience Platform) sorting method, and consistent with the DM25 and Brazil A4 strains (doi: 10.1038/s41598-024-52449-x and 10.1371/journal.ppat.1009254), which are also a TcI strains. To our knowledge, this remains the only chromosome numbering scheme based on a peer-reviewed paper (and with PacBio assembly). The preprint indicated (doi: 10.1101/2025.03.27.645724) has not been published in peer-reviewed journals.

Reviewer: 6. Discussion Lacks Depth. Although the authors have reformulated the manuscript in response to reviewer suggestions, the Discussion section still lacks depth and includes several statements that are either self-evident or overly generic. For example: - “Moreover, MGFs were expressed from various loci across all chromosomes, as detected by RNA-seq and proteomics, indicating no specific genomic site for their expression.” If MGFs are distributed across multiple chromosomes, this observation is expected and does not constitute a novel finding. **Authors:** We included the sentence, lines 282-283, to clarify: “***This is in drastic contrast with other parasites such as *T. brucei* or *Plasmodium sp.* which express their virulent genes from a specific locus.***” We do not understand this outcome as expected considering the biology of these related parasites, and evidence from single-cell RNA-seq (doi: 10.7554/eLife.105822.1) indicating only a few MGF genes are expressed per each *T. cruzi* cell.

- “The MGF genomic organization may facilitate transcription by RNA polymerase II, resulting in constitutive polycistronic transcription and the production of numerous transcripts within a cell.” This reflects a general feature of genome organization in trypanosomatids and is not specific to MGFs. As such, it adds little to the interpretation of the results. **Authors:** We changed the statement, lines 283-286: “***The MGF genes are likely transcribed constitutively, a hallmark of trypanosomatids, resulting in numerous transcripts within a cell, but controlled post-transcriptionally given single-cell analysis suggest only a few MGF genes expressed per cell***⁵³.”

- “However, post-transcriptional levels of expression control may determine which transcripts are translated. This is consistent with the 3’-UTRs of mucins and TSs regulating their expression.” This is a well-established and widely accepted concept in *T. cruzi* gene regulation, and its inclusion here without further analysis or specificity renders it superficial. **Authors:** Statement edited as indicated above.

The discussion would benefit from deeper analysis, a more straightforward integration of the proteomic data with the genomic context, and a more focused examination of what is truly novel or unexpected in the findings. **Authors:** We believe the changes made improved the Discussion focus, clarity, and depth. The Discussion paragraph 2 integrates the proteome and genome data. Paragraphs 3 and 4 integrates the MGF expression changes with yeast surface display screen and significance for parasite infection.

Reviewer: Reframing the focus of the manuscript around proteomic and immunological findings Given the growing number of high-resolution *T. cruzi* genome assemblies—including recent telomere-to-telomere studies—the primary strengths and novelties of this manuscript likely lie in its quantitative proteomics and antigen discovery components. These are valuable datasets that complement the genomic context, providing new insights into stage-specific protein expression and immune recognition. The authors should consider re-centering the manuscript around these contributions, clearly distinguishing between genomic features that are confirmed versus those that are newly discovered, and emphasizing the integrative nature of their approach to parasite biology. In particular, a comparative analysis of the antigenic properties of different MGF members would be a valuable addition, as would a detailed examination of differential expression patterns within each gene family. **Authors:** The changes made in the manuscript as indicated above addresses the reviewer’s suggestions regarding novelty. It is important to note that, besides our work, only two *T. cruzi* genomes at HiFi resolution exist—one still as preprint. This makes our high-resolution genome (the only one performed with PacBio HiFi and Pore-C for the Sylvio X10 strain) a significant advance and important to be detailed. Other relevant aspects of the work, such as proteomics and temporal changes in MGF expression, which are the key findings have also been highlighted.

Reviewer: Recommendation. The manuscript has improved considerably and includes valuable data. However, I recommend minor revision to address the conceptual conflation of MGFs, adopt standard genome terminology, properly contextualize the genomic findings in light of recent literature, and adjust the manuscript’s focus to highlight its most substantial contributions. These changes will ensure the paper is accurate, conceptually rigorous, and impactful within the *T. cruzi* field. **Authors:** All has been addressed above.

Reviewer's comments in black, author's response in blue.

REVIEWERS' COMMENTS

Reviewer #1 (Remarks to the Author):

I acknowledge the authors' efforts in addressing the suggestions, which should contribute to clarifying the work and highlighting the value of the results obtained. In my opinion, this manuscript is now close to being acceptable for publication in Nature Communications.

My only concern, which I consider essential to revise, relates to the transcription of disruptive genes, particularly in the context of the following statements:

Line 282 (added in the last revision): “The MGF genes are likely transcribed constitutively, a hallmark of trypanosomatids, resulting in numerous transcripts within a cell, but controlled post-transcriptionally, given that single-cell analyses suggest only a few MGF genes are expressed per cell.”

Line 302: “This aligns with data showing that most, if not all, TSs are transcribed; (...)”

In reference 52 (Díaz-Viraqué et al.), a deep transcriptome analysis was performed, and the authors reported low or undetectable levels of expression for disruptive genes (TS, MASP, mucins): “By contrast, the disruptive compartment, which in some cases comprises almost entire chromosomes, is enriched in low-expressed genes or genes with undetectable RNA levels (Fig. 2a).”

Authors: For **line 282**, we changed the statement to: “A low MGF gene expression level was detected by transcriptomics at the parasite population level, with some MGF transcripts undetected⁵², further supporting that not all MGFs are expressed. Our TMT-MS did not detect all MGF proteins expressed in the parasite population, implying either transcriptional or post-transcriptional control of their expression.”

For **line 302**, we deleted the sentence. We included in **line 301**: “...However, we cannot rule out differences at the transcriptional level.”

Overall, transcription of disruptive genes is low to very low. Reference 52 further shows that only specific disruptive genes are upregulated in trypomastigotes, and it would be valuable to assess whether this is also reflected at the proteomic level using the data generated in this work. This last point is a suggestion rather than a requirement.

Authors: Figure 3D, F, and E already address RNA-seq and proteomics comparison for core and MGF genes. This is also detailed in Supplementary Figure 10, showing protein expression of core and MGF proteins for each gene in all chromosomes.

What is essential, however, is to avoid referring to the transcription of disruptive genes as constitutive in general terms, since the evidence indicates that their expression is low or undetectable. **Authors:** Done as stated above.